

# System vulnerability and risk assessment of railway systems to flooding
Weihua Zhu[1,2], Kai Liu [1,2*], Ming Wang [1,2], Philip J. Ward[3], Elco E. Koks[3]
*[1] State Key Laboratory of Earth Surface Processes and Resource Ecology, Beijing*
*100875, China*
*[2] Academy of Disaster Reduction and Emergency Management, Faculty of*
*Geographical Science, Beijing Normal University, Beijing 100875, China*
*[3] Institute for Environmental Studies (IVM), Vrije Universiteit Amsterdam, 1081 HV*
*Amsterdam, Netherlands*
Correspondence to Kai Liu (liukai@bnu.edu.cn).
**ABSTRACT:** Floods have negative effects on the reliable operation of transportation
systems. In China alone, floods cause an average of ~1125 hours of railway service
disruptions per year. In this study, we present a simulation framework to analyse the
system vulnerability and risk of the Chinese railway system to floods. To do so, we have
developed a novel methodology for generating flood events at both the national and
river basin scale. The resulting event set provides the basis for national- and provincial-
level railway risk assessments, focusing in particular on affected trains, affected
passengers and increased time for detoured trains. The results show that due to spatial
variations in the railway topology and traffic flows, the system vulnerability of the
Chinese railway system to floods in different basins is highly heterogeneous. Flood



events in the Yangtze River Basin show the largest impact on the national railway system,
with approximately 40% of the national daily trains being affected by a 100-year flood
event in that basin. At the national level, the average number of daily affected trains
and passengers for the national system are approximately 200 trips and 165,000 people
(2.7% and 2.8% of the total daily numbers of trips and passengers), respectively. In
addition, the mean average increased time for detoured trains reaches approximately
five hours. The event-based approach presented in this study shows how we can
identify critical hotspots within a complex network, taking the first steps in developing
climate-resilient infrastructure.
**KEYWORDS**: river basin flooding; railway system; risk assessment; system vulnerability

## 1. Introduction

Floods can have negative effects on transportation systems through both the
destruction of physical infrastructure and the disruption of freight and traffic flows
(Reed, 2004; Moran et al., 2010; Benn, 2013; Kellermann et al., 2015). For example,
during the Tbilisi (Georgia) floods in June 2015, the estimated damage in terms of
replacing affected assets was 14.8 million USD, whilst losses related to increases in
travel time and higher operating costs were estimated at approximately three million
USD (up until autumn 2015) (GFDRR, 2015). In May and June 2013, the Austrian Federal
Railways faced severe damage by the major floods in central Europe, with a total cost
of more than 84 million USD. The event caused extensive damage to track structures


and also caused widespread service disruptions, despite many protective actions that
had been adopted ahead of time (Kellermann et al., 2016). In China, over 2146 rail
service disruption events and over 20,825 hours of discontinued service due to flooding
were reported from 2000 to 2016 (Editorial Board of China Railway Yearbook, 2001-
2017). In 2016, the direct economic loss of the Chinese railway system caused by floods
was approximately 80 million USD (Editorial Board of China Railway Yearbook, 2001-
2017). As such, there is a clear need to evaluate the vulnerability of the transportation
system to extreme flood hazards and to identify high-risk transportation components
to make the transportation systems safer and more effective for operation and
maintenance.
Many studies have investigated flood impacts on transportation systems, focusing on
either flood vulnerability of assets (Kellermann et al., 2015; Pregnolato et al., 2017;
Singh et al., 2018; Koks et al., 2019) or the risk to the entire system (Gil and Steinbach,
2008; Kellermann et al., 2016; Lamb et al., 2019). In these studies, flood vulnerability is
usually defined as the relationship between the characteristics of the transportation
components (i.e., the physical structure, traffic flow and traffic velocity) and the
variables characterizing the intensity of the flood hazard (i.e., flood depth and flood
velocity) (Pregnolato et al., 2017). However, as major river floods are usually driven by
large-scale atmospheric circulations (Prudhomme and Genevier, 2011; Lavers et al.,
2013) and affect large areas, they can disrupt several components concurrently across
a network system (Becker and Grünewald, 2003; Kundzewicz et al., 2013). Within a





network system, the impact on operational performance is often the result of failure of
multiple components in the aftermath of an event (Gong et al., 2017). As such, a
system-level perspective is essential to properly assess transportation system
vulnerability due to flooding.
Some studies have assessed transportation vulnerability to natural hazards from a
system-level perspective (Chang et al., 2010; Hong et al., 2015). Chang et al. (2010)
investigated the potential impacts of climate change on travel disruption in the
metropolitan area of Portland, Oregon. They combined a hydrologic, hydraulic model
and a travel forecast model to process their study. Hong et al. (2015) assessed the
Chinese railway system's vulnerability in terms of traffic flow loss based on historical
flood events from 1981 to 2010. Unfortunately, due to the widespread lack of
appropriate historical flood hazard data and computational issues with running large-
scale hydraulic models (Sene 2008; Chang et al. 2010), research so far has been carried
out only on a case-study basis where historical scenarios are available (Hong et al.,
2015). However, for inter-city and inter-country trade, national and global-scale
transportation systems have flourished in recent decades. Examples include Pan-
European transportation corridors (Janic and Vleugel, 2012) and the railway system of
the Belt and Road Initiative (Yang et al. 2018); therefore, large-scale flood event data
and methods should be improved to assess system-level vulnerability and risk on
operational performance for such large spatial transportation systems.
The recent development of global flood hazard maps (Alfieri et al., 2013; Hirabayashi



et al., 2013; Ward et al., 2013; Sampson et al., 2015;Dottori et al., 2016) has paved the
way for performing large-scale flood risk assessments. These global flood hazard maps
have been widely applied to assess the global risk to flooding in terms of population
(Ward et al. 2013; Arnell et al. 2016; Dottori et al. 2016), gross domestic product (GDP)
(Ward et al., 2013; Winsemius et al., 2013), economic damage (Ward et al., 2013;
Dottori et al., 2016; Winsemius et al., 2016; Ward et al., 2017), and transportation
infrastructure (Koks et al., 2019). Koks et al. (2019), for example, assessed the direct
economic damage to transportation infrastructure assets using a conventional damage
assessment approach through asset-specific fragility curves based on global flood data.
Studies such as these facilitate a better understanding of the impacts of flood hazards
on large-scale transportation systems and provide up-to-date knowledge on risk
analysis frameworks.
This study aims to develop a framework to quantify the system vulnerability and risk
in transportation systems in terms of operational performance loss under large-scale
flood hazards. System vulnerability in this study is represented as the system
performance loss with different flood intensities. Most studies use regional- or national-
scale flood footprints, which show the flood depth for a given return period in that area.
In reality, the presented floods in such as a flood footprint may not all happen at the
same time. When assessing possible cascading effects, the use of independent flood
events is therefore necessary (Nones and Pescaroli, 2016). As such, we develop a
method for generating a set of independent flood events at the national and river basin



scale. Potential performance loss is assessed using network theory and a spatial analysis
method. We illustrate our methodology by applying it to the Chinese railway system.

3        The remainder of the paper is organized as follows. In section 2, we propose a

framework for the evaluation of system vulnerability and risk of flood hazards to
transportation systems and use the Chinese railway system for application, including
how to generate flood events, define the network system for the transportation system,
calculate system vulnerability metrics, and quantify flood risk. Section 3 presents the
main findings and results. Section 4 and Section 5 provide the discussion and conclusion,
respectively, to this article.
## 2. Data and method

11       Flood risk can be defined as a function of flood hazard, exposure and its related

vulnerability. A flood hazard is usually characterised by its intensity and occurrence
probability; exposure refers to the population and assets exposed to flooding; and
vulnerability is often defined as the loss ratio of people or assets suffered to different
intensity of hazard (Gouldby and Samuels, 2009; Haimes, 2009; UNISDR, 2011;
Winsemius et al., 2013). In this work, exposure is represented by the railway network
exposed to the flood hazard. Asset vulnerability is defined as the failure of a railway
asset based on the design standard and is expressed as a failure threshold. If the failure
threshold is exceeded, the service of the component is assumed to be disrupted,
resulting in a 100% performance loss of that asset. System vulnerability is represented



Natural Hazards
and Earth System


as the system performance loss with different flood intensities. Risk is calculated as the
expected annual performance loss at the national and provincial levels.

3       Figure 1 presents an overview of the framework used in this study. First, we generate

a national- and river basin-scale flood event set. To do this, we use flood hazard maps
for different return periods at the national scale, taken from a global flood hazard model.
We then divide these into flood hazard maps for the major river basins and use a curve-
fitting method to estimate the flood depth for any return period for any cell. We then
apply a Monte Carlo sampling method (Metropolis 1987) to generate the flood events
per river basin and aggregate these events to the national scale. Second, we define the
railway system as a network using network theory (Newman, 2010). Third, we intersect
the flood events with the railway network to identify the disrupted segments in the
railway system based on a pre-defined failure threshold. In the last part of our analysis,
we assess the system vulnerability and risk in terms of several performance loss metrics,
including the daily cancelled trains and cancelled passengers, the daily detoured trains
and detoured passengers, the daily affected trains and affected passengers, as well as
the total increased time and the average increased time for the detoured trains. We
also analyse the parameters in the failure threshold sensitivity to the risk result and the
related risk uncertainty.


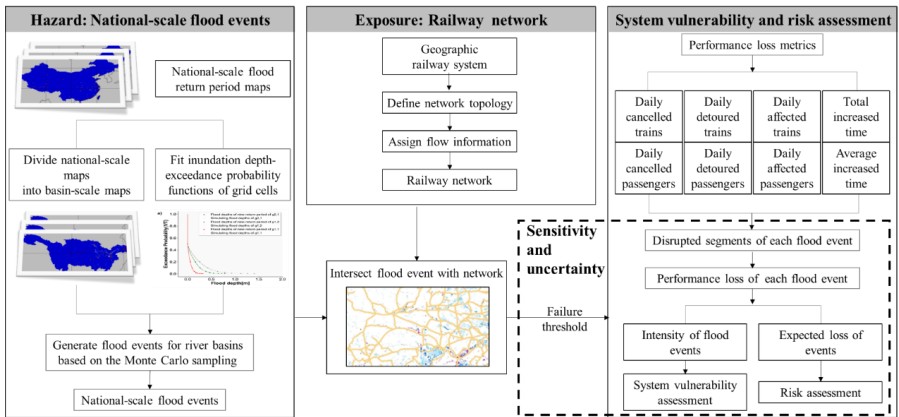

*Fig. 1 Methodology of the flood system vulnerability and risk assessment of railway*
*infrastructure. Railway geometries © OpenStreetMap contributors 2019. Distributed*
*under the Open Data Commons Open Database License (ODbL) v1.0.*

## 2.1 National-scale flood event generation

To ensure the estimation is as accurate as possible for an event-based flood risk
assessment, a large number of independent flood events are required (Speight et al.,
2017; Wu, 2019; Zhu et al., 2020). In this study, we apply a curve-fitting method and a
Monte Carlo sampling method to generate independent flood events using global flood
hazard maps from GLOFRIS for multiple return periods (Ward et al., 2013). In brief, for
each grid cell, we obtain the flood depth from the flood hazard maps for nine different
return periods (2-1000 years). We then fit an inundation depth-exceedance probability
function through these data points, which is used to estimate the flood depths for any
return period. Based on these functions per cell, we apply a Monte Carlo sampling
method to produce basin-specific flood events, which are further combined into a
national independent flood event set (see Section 2.1.3). In this study, we assume that



a flood event within one basin will produce a flood with the same intensity (return
period) within that entire basin, whilst we assume that floods between different basins
are independent of each other (Fraiture, 2007; Rojas et al., 2013). In the following
subsections, we describe the input flood hazard maps, the function fitting procedure,
and the Monte Carlo analysis in more detail.
### 2.1.1 Input flood hazard maps
Our flood hazard data are extracted from the GLOFRIS global fluvial flood hazard
maps of Winsemius et al. (2013), which are developed using the methods provided in
Ward et al. (2013) and Winsemius et al. (2013). The GLOFRIS flood hazard dataset is a
global 30-arcsecond (ca. 1-km) resolution gridded dataset. Hazard maps are provided
for nine return periods (2, 5, 10, 25, 50, 100, 250, 500, and 1000 years). We divide China
into nine major river basins (Fig. 3) according to the main river system from the Data
Center for Resources and Environmental Sciences, Chinese Academy of Sciences, which
is accessible from the Resource and Environment Data Cloud Platform
(http://www.resdc.cn/, last access: 19 May 2020): the Continental Basin, Haihe River
Basin, Huaihe River Basin, Pearl River Basin, Songhua and Liaohe River Basin, Southeast
Basin, Southwest Basin, Yellow River Basin and Yangtze River Basin. As such, we extract
the flood hazard data for each of these river basins.
### 2.1.2 Fitting procedure
For each grid cell, the GLOFRIS maps estimate the flood depth for the nine



aforementioned return periods. To estimate the flood depth for any return period, we
fit a quadratic spline function to develop an inundation depth-exceedance probability
function ($p$) for each grid cell (Marsden, 1974; Vandebogert, 2017; Meshram et al.,
2018). The quadratic spline is a method that uses a piecewise quadratic function to
obtain the best-fitting curves. This interpolation method allows us to obtain a smooth
continuous curve through the provided flood depths for the different return periods.
The method is applied as follows. For each grid cell, the annual exceedance
probability flood depth $D_T$ is calculated by Eq. 1:

$$P(D_T) = \frac{1}{T} \tag{1}$$

where $D_T$ is the magnitude of a flood depth with return period of $T$- year, $P(D_T)$
is the exceedance probability of $D_T$. $D_T$ is between $[D_1, D_{1000}]$, with $D_1 = D_2 \leq$
$D_5 ... \leq D_{1000}$. We assume that $D_1$ is equal to zero (i.e., 1-year event with a flood depth
of 0 m) and is the same as that of a 2-year event (the lowest return period in the
GLOFRIS dataset). Let $Pr(D_T)$ denote a quadratic, continuously differentiable
function of $P(D_T)$. Then, by definition:

$$Pr(D_T) = aD_T{}^2 + bD_T + c \tag{2}$$

For each interval of grid cell $g_{x,y}$, we can obtain its piecewise quadratic function by
Eq. 3:

$$Pr_{x,y}(D_T) = \begin{cases} Pr_{x,y}^1(D_T) = a_1 D_T{}^2 + b_1 D_T + c_1 & D_T \epsilon [D_1, D_{1-T}] \\ Pr_{x,y}^2(D_T) = a_2 D_T{}^2 + b_2 D_T + c_2 & D_T \epsilon [D_{1-T}, D_{2-T}] \\ \quad\quad\quad ... \\ Pr_{x,y}^p(D_T) = a_p D_T{}^2 + b_p D_T + c_p & D_T \epsilon [D_{p-1-T}, D_{p-T}] \end{cases} \tag{3}$$

where $Pr_{x,y}(D_T)$ is a set of continuous inundation depth-exceedance probability




functions    consisting    of    $p$    continuous    quadratic    functions.    For
$a(a_1, a_2, \ldots, a_p), b(b_1, b_2, \ldots, b_p), c(c_1, c_2, \ldots, c_p) \epsilon R$, we can calculate these constants
by bracketing the critical point of $P(D_T)$ and derivative of the function $Pr_{x,y}(D_T)$;
details on the interpolation methods can be found in a previous study by Sun and Yuan
(2006). We assume that only one event occurs per year in each basin. Examples of the
inundation depth-exceedance probability function of grid cells are shown in Fig. 2a.
**2.1.3 Simulation procedure**
To produce a time-series of flood events based on the created inundation depth-
exceedance probability functions (Section 2.1.2), we use a Monte Carlo sampling
method. The basic idea of the Monte Carlo sampling method is that when the number
of simulations is sufficiently large, the frequency of an event approximates the
probability of the occurrence of the event (Baker, 2008; Speight et al., 2017). The flood
event generation procedure is presented in Fig. 2 and Appendix Fig. A1 and can be
summarized in two steps. First, we generate a set of independent events at the basin
scale. For each event $E_j^i$, and for each basin $B_j$, a random number $P_j^i$ between 0 and
1 is generated from a uniform distribution. The flood depth of the cells in basin   for
event $E_j^i$ can be calculated using $P_j^i$ and the inundation depth-exceedance
probability function based on the assumption that a flood event in one basin will
produce a flood with the same intensity. This concept is presented in Fig. 2a-b. Second,
we generate a set of national-scale independent flood events. For a national-scale flood



event, basin-specific floods of nine basins can be randomly combined into a national-
scale flood by assuming independence between the flood events among different
basins, as presented in Fig. 2c-d.

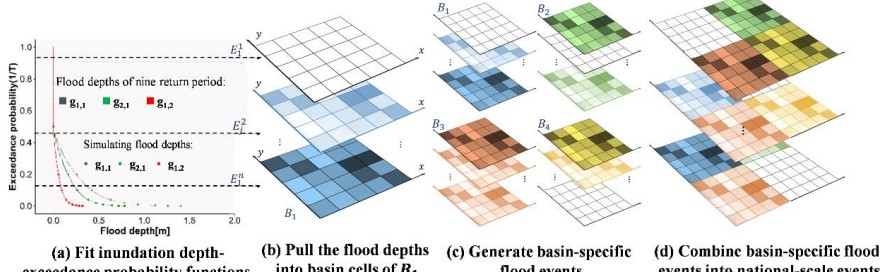

(a) Fit inundation depth-          (b) Pull the flood depths      (c) Generate basin-specific     (d) Combine basin-specific flood
exceedance probability functions   into basin cells of $B_1$       flood events                    events into national-scale events

*Fig. 2 A flowchart of national-scale flood event generation. $E_1^1$  (the top layer in (b)) is a*
*flood event in basin $B_1$, where $P_1^1 = 0.93$; $E_1^2$  (the middle layer in (b)) is a flood event in*
*basin $B_1$, where $P_1^2 = 0.45$; and $E_1^3$  (the lower layer in (b)) is a flood event in basin $B_1$,*
*where $P_1^3 = 0.17$. The upper-left layers in (c) are a basin-scale flood set of basin $B_1$, the*
*upper-right layers in (c) are a basin-scale flood set of basin $B_2$, the lower-left layers in (c)*
*are a basin-scale flood set of basin $B_3$, and the lower-right layers in (c) are a basin-scale*
*flood set of basin $B_4$. The top layer in (d) is combined with the top four layers in (c) (the*
*four basin-scale floods of basin $B_1, B_2, B_3$  and $B_4$  combine into a national-scale flood*

13                                       *event).*

For each year, we assume that within each basin, only one flood event can occur. For
each basin to obtain 10,000-year events (we assume that 10,000-year of events are
sufficient to cover almost all probable scenarios), we therefore apply a Monte Carlo
method to sample 10,000 exceedance probabilities. For each of these exceedance
probabilities, we estimate the inundation depth for each cell within that basin (i.e.,
assuming that the exceedance probability is the same throughout the entire basin). We



repeat this procedure for each basin, which results in a 10,000-year set of flood events
for each basin. We then combine these sets into a national scale flood event set by
assuming independence between the flood events in the different river basins (Fig. 2d).
Hence, for each of the 10,000 years, we simply take the estimated flood depths for each
basin. For example, in year 1, basin 1 may have an exceedance probability of 0.5, whilst
basin 2 may have an exceedance probability of 0.98. For year 1, the resulting national-
scale flood map would therefore have values for a flood event with an exceedance
probability of 0.5 in basin 1, a flood event with an exceedance probability of 0.98 in
basin 2, and so forth. This procedure results in a 10,000-year national-scale flood event
set.
We also assess the system vulnerability by calculating the impacts that could occur
throughout China if a flood with a given return period were to occur within an individual
basin. To do this, for each basin and each return period we draw 10,000 events for all
other basins assuming independence. In total, this leads to a set of 810,000 events
(10,000 events x 9 return periods x 9 basins).
## 2.2 Railway network building
Railway systems are commonly represented through spatially explicit networks as an
analogy for their structure and flows (Rodrigue, 2016). This network representation can
be used to calculate system performance metrics based on network theory. In this work,
the Chinese railway system was modelled as a directed weighted network, which





consists of a group of nodes (stations) and connected by edges (railway lines) with daily
train trips, where the edges have a travel direction associated with them. To build the
Chinese railway network, we use the geographic information of railway system from
OpenStreetMap (OSM) and the timetable data including daily number of trains and
associated routes from the Railway Service Website (Liu et al., 2018; Zhu et al., 2020).
As our method is primarily concerned with flood risk along rail segments between cities
and not within cities, for simplicity, we combine multi-stations into one node using the
location of the highest capacity station in each city. In total, 2240 nodes are combined
into 1790 nodes. The final extracted railway network has a total length of 90,600 km for
(merged parallel) lines connecting two identical stations, consisting of 1973 edges and
1790 nodes (Fig. 3). Figure 3 shows the spatial distribution of the railway network and
average daily numbers of trains. Topology and traffic flows vary greatly in spatial apace.
The network density, reduces greatly moving from Eastern China to Western China. For
the traffic flow, the railways connect large cities, like the railways from Beijing to
Guangzhou, Harbin and Shanghai, and railway from Shanghai to Changsha have higher
flows.

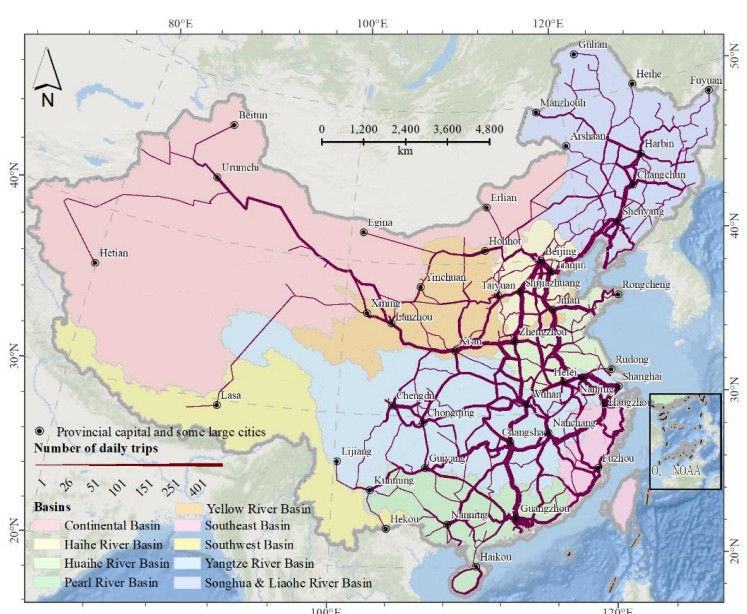

Fig. 3 The spatial distribution of the railway network and average daily numbers of trains.

The river basins layer comes from the Data Center for Resources and Environmental

Sciences, Chinese Academy of Sciences, which is accessible from the Resource and

Environment Data Cloud Platform (http://www.resdc.cn/, last access: 19 May 2020).

Railway geometries © OpenStreetMap contributors 2019. Distributed under the Open

Data Commons Open Database License (ODbL) v1.0. The timetable data included the

daily number of trains and associated routes from the Railway Service Website (Liu et al.,

2018).

## 2.3 Failure condition based on an event

We assume that a railway is impassable when the water level on the railway line is

higher than the failure threshold $Wd$ of the railway service after drainage (CRPH, 2012;

Espinet et al., 2018). The water level after drainage $WL_{x,y}$ of grid cell $g_{x,y}$ is

calculated by Eq. 4:



$$WL_{x,y} = D_{T\,x,y} - Wld_{x,y} * Dc \qquad (4)$$
where $D_{T\,x,y}$ is the flood depth of a flood event, $Wld_{x,y}$ is the water level of the
design standard (i.e., the return period t) of grid cell $g_{x,y}$, and $Dc$ is the drainage
capacity rate.
The rail segment $l_{ij}$ between two stations failure condition is defined by Equations
5 and 6:
$$Fc_{ij} = \prod_{xy}^{ij} Z(xy) \qquad (5)$$
$$Z(xy) = \begin{cases} 0, WL_{x,y} \geq Wd \\ 1, WL_{x,y} < Wd \end{cases} \qquad (6)$$
$Fc_{ij}$ is the failure condition of component $l_{ij}$, which has two states, namely, normal
(denoted by 1) and disrupted (denoted by 0), resulting in 100% disruption. $Z(xy)$ is
the failure condition of grid cell $g_{x,y}$; when the water level after drainage is larger than
$Wd$, $Z(xy)$=0; otherwise, $Z(xy)$=1.
In this study, we consider a failure threshold of 0.2 m after drainage, according to the
railway transportation emergency plan (CRPH, 2012; Espinet et al., 2018). The flood
design standard of the culverts, bridge and embankments of the Chinese national
railway system is designed for 100 year in China, according to the standard for flood
control (CRPH, 2016). Furthermore, we assume that the drainage capacity rate is 0.8 of
water level of the design standard, and it reduces the total amount of water that the
railway structure can actually drain (TB 10001, 2016; Espinet et al., 2018).
Failure hotspots of railway segments $l_{ij}$ can be found by the annual failure
probability $AF_{ij}$, which is calculated by Eq. 7:



$$AF_{ij} = \frac{\sum_e^E FC_{ij}^e}{N} \qquad (7)$$
where $AF_{ij}$ is the failure probability to the railway segments, $E$ is the N-year flood
events catalogue, and $FC_{ij}^e$ is the failure condition of railway segment $l_{ij}$ under flood
event $e$.

## 2.4 Calculating system vulnerability and risk

### 2.4.1 Performance loss metrics

*Daily affected trains and the associated daily affected passengers*
Once a flood occurs, trains may be affected in two ways: (i) increased travel time; or
(ii) cancellation. The number of daily affected trains $N_e^{tol}$ is calculated by Eq. 8:
$$N_e^{tol} = N_e^c + N_e^d \qquad (8)$$
Where $N_e^c$ the number of daily is cancelled trains and $N_e^d$ is the number of daily
detoured trains after a flood event.
We assume that the average number of passengers is 80% of the train's capacity (Wei
et al., 2017)(Rezvani et al., 2015)(Rezvani et al., 2015)(Rezvani et al., 2015)(Rezvani et
al., 2015). As such, the number of affected passengers $P_e^{tol}$ can be defined by Eq. 9:
$$P_e^{tol} = \sum_i^{(N_e^c+N_e^d)} CA_i * 0.8 \qquad (9)$$
where $CA_i$ is the capacity of the $ith$ train.
*Daily detoured trains and the associated daily detoured passengers*
Once a flood occurs, some trains will detour to complete their journeys. The daily
detoured trains $N_e^d$ can be calculated based on four assumptions as follows (in order



of descending priority), which is also presented in Appendix Fig. A2:
① Stations are not repeated along the routes;
② The train passes the largest number of original stations along the detoured

4        route;

③ The detour with the smallest increase in travel time is selected;
④ Detouring is impossible when the increased time for re-routing is greater than

7        24 hours.

the daily detoured passengers $P_e^d$ can be defined by Eq. 10:
$$P_e^d = \sum_i^{(N_e^d)} CA_i * 0.8 \qquad (10)$$
where $N_d$ is the daily detoured trains and $CA_i$ is the capacity of the $ith$ train.
***Total increased time for the detoured trains***
The total increased time $T_e^{tol}$ for detoured trains is calculated by Eq. 11:
$$T_e^{tol} = \sum_i^{N_d} T_i^e - \sum_i^{N_d} T_i \qquad (11)$$
where $T_i^e$ is the running time of the $ith$ train under flood event e, and $T_i$ is the
original travelling time of the $ith$ train.
***Average increased time for the detoured trains***
The average increased time is calculated by Eq. 12:
$$T_e^{ave} = \frac{T_e^{tol}}{N_e^d} \qquad (12)$$
where $T_e^{ave}$ is the average increased time under flood event e and $N_d$ is the
number of detoured trains.



*Daily cancelled trains and the associated daily cancelled passengers*
Once a flood occurs, some trains may be cancelled if there is no alternative route
possible or when the re-routing time is too long (greater than 24 hours). The daily
cancelled trains $N_e^c$ is calculated by Eq. 13:

$$N_e^c = N_S - N_e^s \tag{13}$$

where $N_e^c$ is the daily cancelled trains after a flood event, $N_e^s$ is the number of
running trains in the system after a flood event, and $N_S$ is the original number of trains
in the system.
Daily cancelled passengers $P_e^c$ can be defined by Eq. 14:

$$P_e^c = \sum_i^{(N_e^C)} CA_i * 0.8 \tag{14}$$

where $N_c$ is the daily cancelled trains and $CA_i$ is the capacity of the $ith$ train.
2.4.2 Calculating system vulnerability and risk
Each performance loss metric is calculated for each flood event. System vulnerability
curves are generated to present the relationship between performance loss and flood
intensity (return period). We use the expected daily affected trains, cancelled trains,
detoured trains, affected passengers and increased time for detoured trains to present
the flood risk to the railway system according to Eq. 15:

$$AR_s = \frac{\sum_e^E V_e}{N} \tag{15}$$

where $AR_s$ is the expected daily flood risk level to the railway system, $E$ is the N-
event    flood    catalogue,    and    $V_e$    is    the    performance    loss    metric,





i.e., $N_e^d$, $N_e^c$, $N_e^{tol}$, $P_e^d$, $P_e^c$, $P_e^{tol}$, $T_e^{tol}$, and $T_e^{ave}$  under flood event  $e$, which is defined in
Eqs. 9-14.
## 2.5 Uncertainty and sensitivity analysis
By applying an uncertainty analysis (UA), we identified the range of model output for
imprecisely known input parameters (De Moel et al., 2012). A sensitivity analysis (SA)
aims to determine the parameter effect on the model output (Koks and Haer, 2020).
Parameters with greater effect should attract more additional attention to deal with the
uncertainty they bring (Koks and Haer, 2020; De Moel et al., 2012). Detailed methods
of uncertainty and sensitivity analysis can be found in previous studies by De Moel
(2011) and Koks and Haer (2020).
In this study, we make assumptions on the train disruption threshold using three
parameters (the water level failure threshold, drainage capacity rate, and design
standard) based on emergency code and design code standards (CRPH 2012). However,
it should be noted that these standards are not known exactly for each asset and will
change over time, such as dynamically changing protection standards and ageing
infrastructure. Within a railway system, a lot of different asset types exist, with varying
design standards. This implies that the capacity to cope with the hazard does vary from
location to location. As such, it is worthwhile to perform a sensitivity analysis on these
key parameters (De Moel and Aerts, 2011; Horacio et al., 2019). Hence, we perform an
uncertainty and global sensitivity analysis in which we assess the performance loss


metrics for a range of different values for these parameters. For water level failure, we
use a range between 0.1 and 0.5 m. For the drainage capacity rate, we use a range
between 0.7 and 0.9, and for the design standards, we use a range between 50 and 100
years. The list of all assumptions taken in this study and their range in the sensitivity
analysis can be found in appendix. In total, we create a set of 1000 different parameter
value combinations in the sample space.
## 3 Results
### 3.1 Failure hotspots of railway segments
The annual failure probability of the network segments is shown in Fig. 4 and is
calculated based on the 10,000-year national flood event set. The results show a clear
regional differentiation (Fig. 4a). Areas with high annual failure probabilities are mainly
located in the Yangtze River Basin, Southeast Basin, and Pearl River Basin areas. These
three basins have a humid subtropical climate and high precipitation levels in the rainy
season during the summer; and these areas also have the highest railway density (Fig.
3), mostly across rivers and located on flat area in China, which makes these railway
lines susceptible to flood hazards.
Figure 4b shows the percentage of the length of railway lines that fall into each failure
probability category for the national- and basin-level analyses. Nationally, the failure
probability is greater than 0 for more than 55% of the total length of the railway lines.
This percentage is heterogeneous across different river basins: it is highest in the

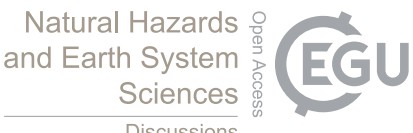
Southeast Basin, followed by the Pearl River Basin and the Yangtze River Basin.
Nationally, 6.8% of the length of the railway lines has a failure probability greater than
0.02, with the highest proportions in the Yangtze River, Yellow River, and Southeast
Basins, with 12.5%, 10% and 7.2%, respectively.

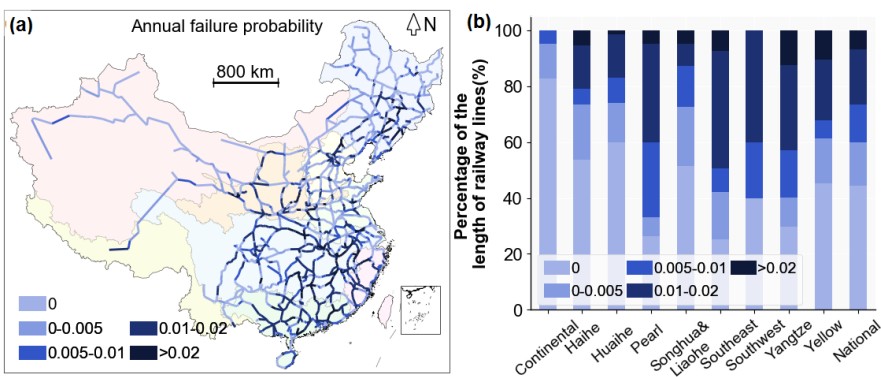

*Fig. 4 (a) Annual failure probability map of the network segments affected by floods and*
*(b) the percentage of the length of railway lines for different failure probability categories*
*per river basin. Railway geometries © OpenStreetMap contributors 2019. Distributed*
*under the Open Data Commons Open Database License (ODbL) v1.0.*
## 3.2 Risk analysis of the Chinese railway system
The performance loss distribution curves of the railway system using the 10,000-year
national-scale flood set are presented in Fig. 5. The results show that approximately 85%
of the flood events have little effect (less than 0.01 of the daily trains and passengers) on
the railway system from the perspective of all the performance metrics. For the daily
affected trains, the absolute maximum number can reach 4200, and the average number
is approximately 200 trips; these values represent 59% and 2.7% of the number of the





daily trains. For the daily affected passengers, the absolute maximum number can reach
3,500,000, and the average number is approximately 165,000 people (60% and 2.8 of the
number of the daily passengers). In addition, the largest average increased time for detoured
trains can reach 14 hours and the mean average increased time for detoured trains is
approximately 5 hours.

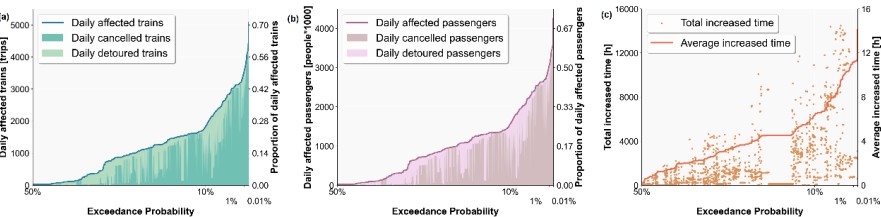

*Fig. 5 Exceedance probability-performance loss curves*
The performance losses per province of the railway system are presented in Fig. 6 for
a range of metrics. The risk differs considerably between regions when expressed in
different risk metrics. When examining the metrics of the daily affected trains and
affected passengers, we find that the provinces in Central China, such as Henan, Hubei
and Anhui, have the highest absolute and relative risks, estimated to be over 40 daily
affected trains (4.5% relative to the number of the province's daily trains) and more
than 35,000 daily affected passengers (3.5% relative to the number of the province's
daily passengers). Interestingly, some provinces, such as Tibet Province, have a low risk
in absolute terms but a high risk in relative terms because the Tibet Province has the
smallest rail network and rail traffic density; only one line (i.e., Qinghai-Tibet Railway)
crosses this region, which is therefore highly vulnerable to even a low-frequency flood

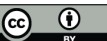

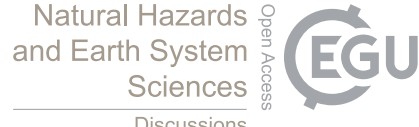

hazard. Guangdong Province has the opposite results, with high risk in absolute terms
and low risk in relative terms due to the large rail network and rail traffic density, which
make the railway system more robust even with a high flood failure probability. The total
and average increased time for detoured trains show contrasting results. The high risk
in terms of the total increased time is mostly distributed in East China, whereas the
highest average increased time is distributed in western provinces such as Xinjiang and
Tibet Provinces. From Eastern China to Western China, the traffic flow becomes
significantly lower; more trains can be detoured with less time per trip in East China,
and in the western provinces, fewer trains can be detoured but with more time per trip.

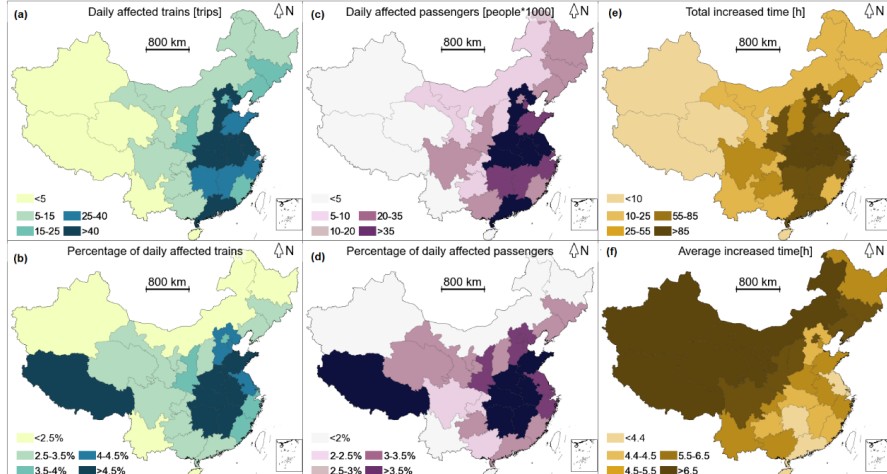

*Fig. 6 Performance loss of the railway system per province. (a) The daily affected trains*
*in absolute terms; (b) the daily affected trains relative to the number of the province's*
*daily trains; (c) the daily affected passengers in absolute terms; (d) the daily affected*
*passengers relative to the number of the province's daily passengers; (e) the daily total*
*increased time for the detoured trains per province; and (f) daily average increased time*
*for the detoured trains per province. Appendix Fig. A3 provides the risk map of detoured*





*and cancelled trains and detoured and cancelled passengers. Appendix Fig. A4 provides*

*a map of the Chinese provinces*

Several provinces appear at the highest level of the three metrics presented in Fig. 6

and can be classified as particularly vulnerable provinces. Anhui Province, for example,

has one of the highest absolute and relative levels of risk to trains and passengers in Fig.

6a-d but also has the highest total increased time in Fig. 6e. Hubei Province shows one

of the highest absolute and relative levels of risk to trains and passengers in Fig. 6a-d.

Jiangsu Province has the highest absolute levels of risk to trains and passengers in Fig.

6a and c and one of the highest total increased time in Fig. 6e. These provinces are at

the highest risk compared to the other provinces.

## 3.3 System vulnerability of the Chinese railway system

Figure 7 presents system vulnerability curves based on the 810,000 simulated flood

events and shows the performance loss metrics (namely, the percentage of daily

affected trains and increased time) plotted against the return periods. The bottom-right

plots for subfigures a and b show the national results, whilst the other figures show the

results for each river basin. The colour shade represents the distribution of the flood

performance loss, where the lines refer to the median performance loss value and the

bounded lines refer to the 10th and 90th percentiles. The low-impact events cause the

median values to be the same as the lower bound for the nine river basins as a result

of their high frequency.

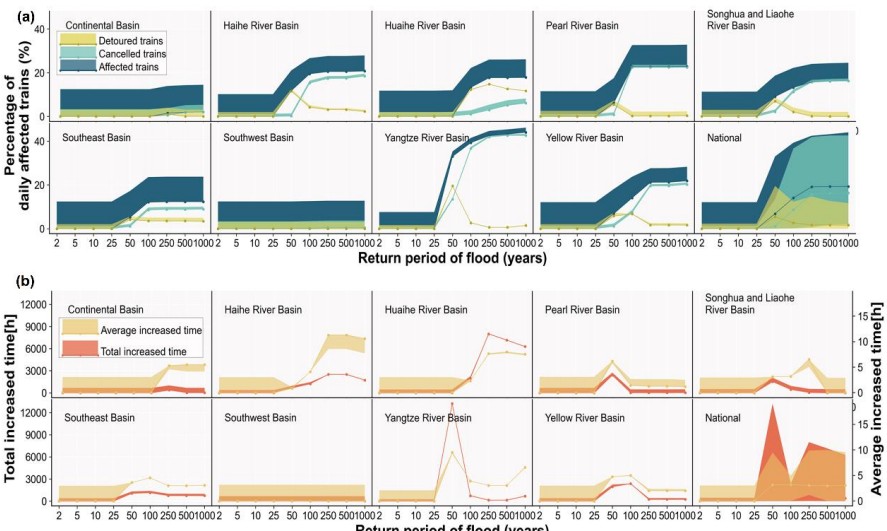

*Fig. 7 System vulnerability curves induced by river floods from the national flood event*

*set, showing: (a) the percentage of daily affected trains to the total number of daily trains*

*and; (b) the increased time for the detoured trains. The shading shows the distribution of*

*the flood performance loss, where the lines refer to the median performance loss value*

*and the bounded lines refer to the 10th and 90th percentiles. In the Appendix Fig. A5, we*

*provide the system vulnerability curves for the passenger-level metrics. NB: for total*

*increased travel time, the values can decrease at higher return periods – this is because*

*some of the trains are cancelled and therefore there is no travel time for those trains*

Due to the different definitions and focus of each metric, the relationship between

each metric and flood intensity is also different. From Fig. 7a, we can see that the

percentage of daily affected trains and daily cancelled trains to the total number of daily

trains increases with the increases of the return period of the flood events for the nine

basins. The percentage of daily detoured trains to the total number of daily trains and

the total and average increased time for detoured trains do not always increase with



increasing return period shown in Fig. 7a and Fig. 7b. The median performance loss for
the five metrics is close to zero for floods with a return period below 25-years and
remains stable when the flood hazard return period exceeds 100-years because of the
railway design protection standards and assumed drainage capacity. Between the 25-
year and 100-year flood events, the percentage of daily affected trains and daily
cancelled trains relative to the total number of daily trains per flood event increases.
The percentage of daily detoured trains relative to total daily trains and the total and
average increased time, increases between the 25-year and 50-year flood events, and
sharply decreases between the 50-year and 100-year events, especially for the Yangtze
River, Yellow River and Pear River Basin floods. This is because most of the north-south
rail lines in China, such as the Beijing-Guangzhou and Beijing-Jiulong lines, cross these
basins. Most trains that are detoured for a 50-year event cannot be detoured for a 100-
year event, as most of the north-south rail lines suffer failures at this hazard intensity.

14       When comparing the results between the nine river basins, we find that, in general,

floods in the basins in central and eastern China have the highest impacts on the
Chinese national railway system. The percentage of daily affected trains (cancelled and
detoured trains) of the total number of trains is the largest for the Yangtze River Basin,
followed by the Pearl River Basin and the Yellow River Basin. In the Yangtze River Basin,
the median percentage of daily affected trains (cancelled and detoured trains) to the
total number of trains is close to 40% for a 100-year flood event. For the Continental
and Southwest Basins, the value is close to zero. The high impacts of daily affected trains





observed in the central and eastern area are due to a significantly higher railway line
density and daily train flows compared to the more inland river basins (see Fig. 3). The
higher annual failure probability of the rail segments in the central and eastern regions
shown in Fig. 4 also causes a large chance of failed railway segments per flood event
and results in higher impact. The daily detoured trains in the Huaihe and Haihe River
Basins in eastern China are higher compared to other basins, which leads to a large total
increased time when one flood occurs. The reason is that the Huaihe and Haihe River
Basins are located in eastern China and only cross railway lines in the eastern coastal
area; therefore, the affected trains have more detour options through the lines of the
Yangtze and Yellow River Basins, which lead to more detoured trains and associated
total increased time.
## 3.4 Risk uncertainty and parameters sensitivity
Figure 8 and Appendix Fig. A6 present the sensitivity of the results to the assumed
parameters and the range of performance metric uncertainty. Overall, from the
uncertainty histograms, we can see that all the performance metrics are right-skewed,
especially for the average daily affected and affected passengers shown in Fig. 8a and c,
and average daily cancelled trains and cancelled passengers shown in Appendix Fig. A6b
and d, they have a long right tail for high performance loss estimates. This seems a little
bit less for the average daily detoured trains and passengers showed in Appendix Fig.
A6a and c, and average increased time for detoured trains showed in Fig. 8e, which is
probably the result of the assumption that detouring is impossible when the increased
time for re-routing is greater than 24 hours, resulting in a smaller range of detoured
options and thus a smaller range in resulting performance loss estimates. The average
number of daily affected trains ranges from 100 to 500 trips; for daily affected
passengers, it is range from 100,000 to 450,000 people, and the average increased time
ranges from 3.5 hours to 5.5 hours with the change in the parameters. The results show
that the performance loss estimates are particularly sensitive to the values used for the
design standards; using the different parameter settings, we see a variation in the
design standards of approximately 43%. The variation in the drainage capacity rate and
water level threshold produces similar uncertainty as the capacity loss, which is
approximately 28%.

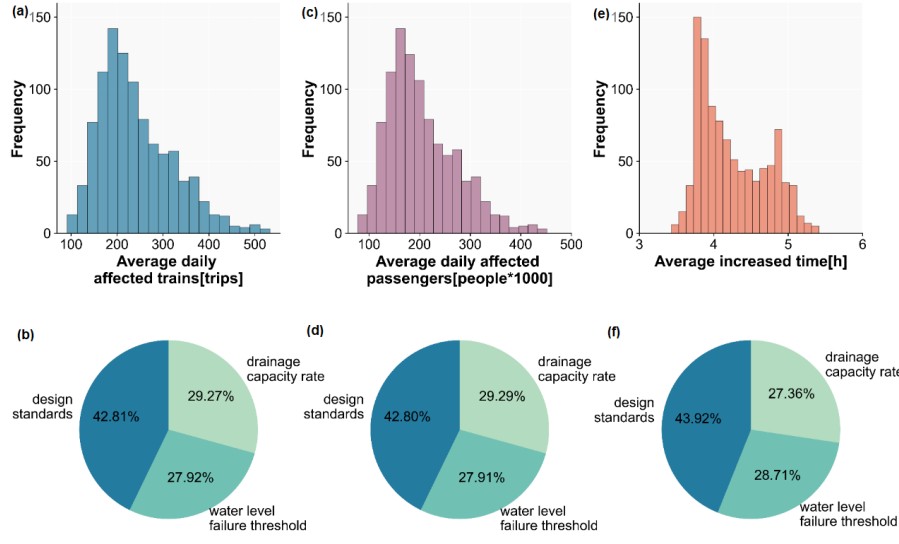

*Fig. 8 Results of the uncertainty (histograms) and sensitivity (pie charts) analyses for the*
*performance metrics. (a) and (b) average daily affected trains; (c) and (d) average daily*





*affected passengers; (d) and (f) average increased time. Fig. A5 provides the results of*
*the other performance metrics.*

## 4 Discussion

Our results reveal clear geographical disparities in the failure hotspots. Areas with
high annual failure probabilities are mainly located in the Yangtze River Basin, Southeast
Basin, and Pearl River Basin. Comparing the failure probability from this study with the
susceptibility map presented in seminal works by Liu et al. (2018a, 2018b), we find some
differences in hotspots in Xinjiang Province and along the Beijing-Shanghai line. In our
study, we find lower failure probabilities relative to the work of Liu et al. For other
regions, the spatial patterns are similar. Our study considers the same protection
standards (the water level failure threshold, drainage capacity rate, and design standard)
for the railway lines in the Chinese railway system. It should be noted that these
standards will not remain constant over time, as a result of ageing infrastructure. This
means that the failure probability in some areas in this study is biased compared to
research based on historical data. Indeed, many older lines have been
upgraded/improved so that the protection standards are more consistent with newer
lines.
In our work, we find that in the Yangtze River Basin, the median relative cancelled
trains to total daily trains is between 0 and 14% when the flood intensity is between 25
and 50-year. In 2016, from May to July, the Yangtze River Basin and Huaihe River Basin
suffered by severely rainfall (Lyu et al., 2018). In most affected areas within the Yangtze





River Basin, the floods that occurred exceeded the 25-year return period. Floods caused
disruptions on several railway lines, including the Chengdu-Chongqing line, Hefei-
Jiujiang line, and Sichuan-Guizhou line, that cross the Yangtze River Basin. In the Huaihe
River Basin, damage occurred to the Beijing-Guangzhou line. From 30 June to 6 July,
approximately 100 trips (approximately 2% of the daily trains) were cancelled every day
for the Chinese railway system. These observed impacts are within the range of our
estimates.

8        In this study, we assumed that within a river basin, the flood probability is constant,

whilst among different basins it is fully independent. In future work, we will assess the
dependence structure of flood hazards within and between basins, for example, by
means of the copula approach as presented in (Jongman et al., 2014). As we assumed
a disruption time of one day due to the lack of information on flood duration in this
study, we may have underestimated the operational performance losses.

14       Using our current approach, the performance loss can be used as the start of the

indirect risk assessment from the travel journey perspective. By combining the ticket
prices and the operating cost per kilometre, the economic loss for the railway company
can be calculated based on the affected trains and associated passengers (Lamb et al.,
2019). As a key mode of transport for interregional trade, the failure of railway systems
can produce large shocks for industries dependent on the supply that may come from
flooded businesses. The risk values per province (such as expected daily cancelled trains
and passengers) can be used as indicators to link with business disruptions. Future work





can try to assess the interregional trade based on the Input and Output table and
regional railway transportation performance decreased in this work. The assessment of
shocks and indirect economic losses induced by railway system failures is essential for
policymakers to design railway infrastructures and to measure indirect economic losses.

## 5   Conclusion

The increased frequency of extreme flood events, coupled with interregional trade
growth, requires national- and global-scale transportation networks to be more resilient
to cope with disruptive events. Evaluation of system-level vulnerability and
identification of risk hotspots is a first step to enhance the robustness of the transport
system. This study presents a framework for performing system-level vulnerability and
risk assessments of a railway system under flooding. The developed framework couples
simulated flood events with state-of-the-art network analysis to measure system
disruptions caused by floods to identify risk hotspots. This work quantifies the system
vulnerability and risk in terms of the performance loss of the Chinese railway system,
induced by the flooding. Results show that failure hotspots, system vulnerability and
the risk of the Chinese railway system under floods are highly heterogeneous. In
addition, the adopted vulnerability metrics present different results in terms of the
system vulnerability and risk.
High failure hotspots are mainly distributed in South China, i.e. Yangtze River, Pearl
River and Southeast Basins. The humid subtropical climate and severe flood hazards in


these areas result in large chances of disruption. For the system vulnerability, the
heterogeneity is largely due to a spatially imbalanced railway topology and traffic flow
as well as a spatially heterogeneous hazard intensity distribution among China. In
general, floods in the basins in central and eastern China have the highest impacts on
the Chinese railway system. Floods in the Yangtze River Basin have the largest impact
on the daily cancelled trains and associated daily cancelled passengers. In the Yangtze
River Basin, the median percentage of daily affected trains to the total number trains
can reach to 45% for a 1000-year flood event. In addition, floods in the Huaihe and
Haihe River Basins cause the largest number of the detoured trains as well as associated
increased time for the Chinese railway system compared with other basins. Finally, this
work quantifies the performance risk due to flooding at the national and provincial level.
We find that, at a national level, the average daily number of affected trains and
passengers are approximately 200 trips and 165,000 people (2.7% and 2.8% of the total
daily numbers of trains and passengers), respectively. The mean average increased time
for detoured trains reaches approximately 5 hours. At the provincial level, the provinces
in Central China have the highest absolute and relative risks, estimated to be over 40
daily affected trains (4.5% relative to the number of the province's daily trains) and
more than 35,000 daily affected passengers (3.5% relative to the number of the
province's daily passengers). The high risk in terms of the total increased time is mostly
distributed in East China, whereas the highest average increased time is distributed in
western provinces, such as Xinjiang and Tibet Provinces. The developed system



vulnerability curves and flood risk maps can provide the information for the decisions
on safety and effectiveness of operation and maintenance. Various performance
metrics can be considered by management departments based on their particular
problems.
**Code/Data availability**
Supporting data are accessible through the associated reference. The data in this study
were analyzed with Python package, and the figures were created with ArcViewTM GIS
and Python packages. All codes used in this work are available upon request.
**Author contribution**
Kai Liu and Weihua Zhu developed the original idea and designed the analyses. Philip
Ward and Elco Koks contributed to the study design. Weihua Zhu, Kai Liu and Elco Koks
conducted the analysis. Weihua Zhu wrote the original manuscript, and Kai Liu, Ming
Wang, Philip Ward and Elco Koks provided comments and revised the manuscript. All
the coauthors contributed to scientific interpretations of the results.
**Declaration of Competing Interest**
The authors declare that they have no known competing financial interests or personal
relationships that could have appeared to influence the work reported in this paper.
**Acknowledgments**
This work was supported by the National Key Research and Development Plan [grant





number 2018YFC1508802]; the National Natural Science Foundation of China [grant
number 41771538]; and PJW received funding from the Dutch Research Council (NWO),
in the form of a VIDI grant [grant number 016.161.324]. EEK received funding from the
Dutch Research Council (NWO), in the form of a VENI grant [grant number
VI.Veni.194.033]. The financial support is highly appreciated.

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



1 Appendix

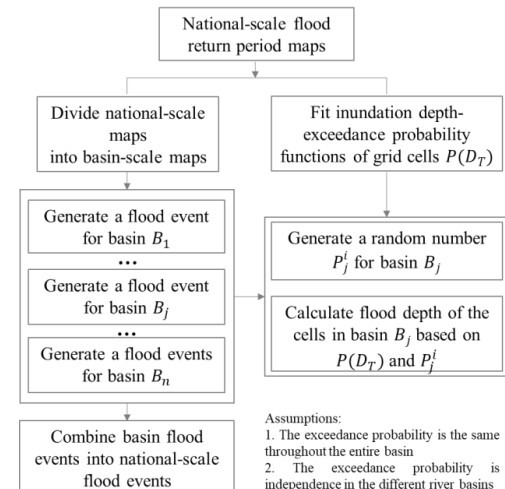

3 *Fig.A1 A flowchart to generate flood event*

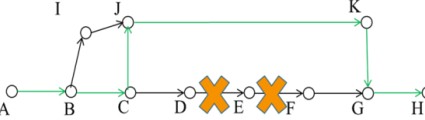

Railway network
Network nodes: $A - K$
Network edges: $AB, ..., KG$
Train trips information:
Trip1: $A \rightarrow B \rightarrow C \rightarrow D \rightarrow E \rightarrow F \rightarrow G \rightarrow H$(passed and stopped stations)
DE and EF are disrupted by the flood event.

Two routes can complete the detour:
A-B-I-J-K-G-H
A-B-C-J-K-G-H
Based on the 'Pass the most original stations', the green routes have
Been chose for detour.

5 *Fig. A2 An example for detour*

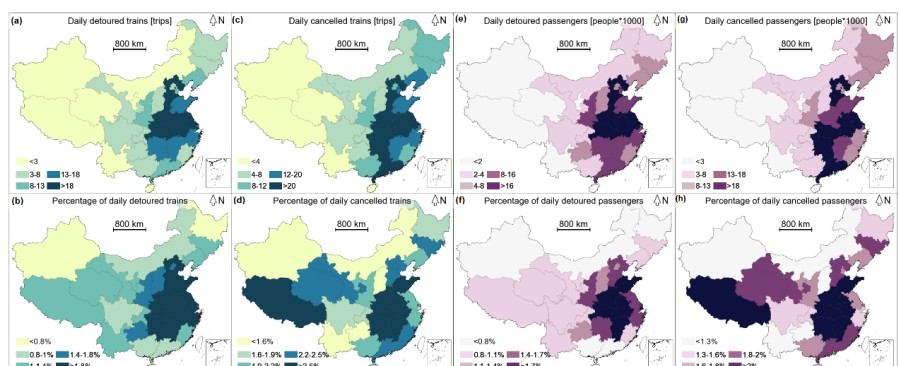

*Fig. A3 Performance loss of the railway system per province. (a) presents the daily detoured trains in absolute terms; (b) presents the daily detoured trains relative to the number of the province's daily trains; (c) presents the daily cancelled trains in absolute terms; (d) presents the daily cancelled trains relative to the number of the province's daily trains; (e) presents the daily detoured passengers in absolute terms; (f) presents the daily detoured passengers relative to the number of the province's daily trains; (g) presents the daily cancelled passengers in absolute terms; (h) presents the daily cancelled passengers relative to the number of the province's daily trains.*

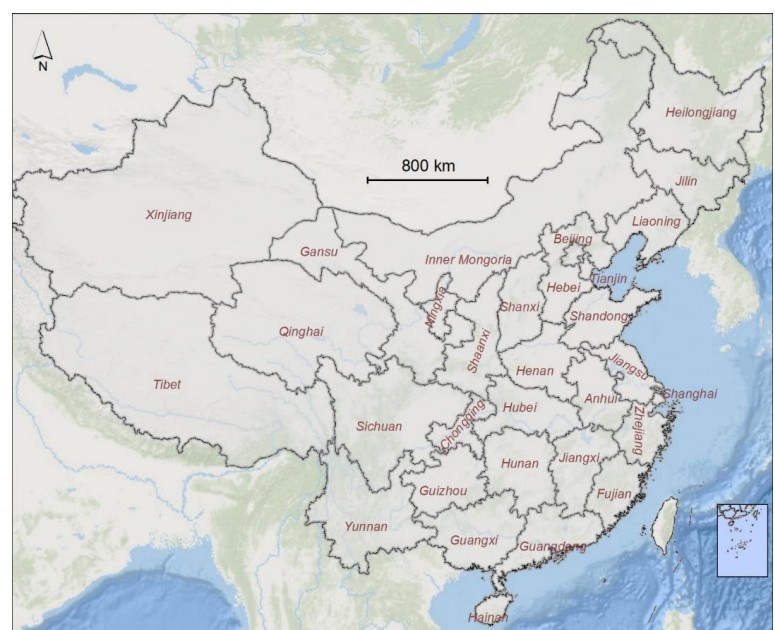

*Fig. A4 Chinese provinces distribution map. The China Provincial Map layer comes from*
*the Data Center for Resources and Environmental Sciences, Chinese Academy of*
*Sciences, which is accessible from the Resource and Environment Data Cloud Platform*
*(http://www.resdc.cn/, last access: 19 May 2020).*

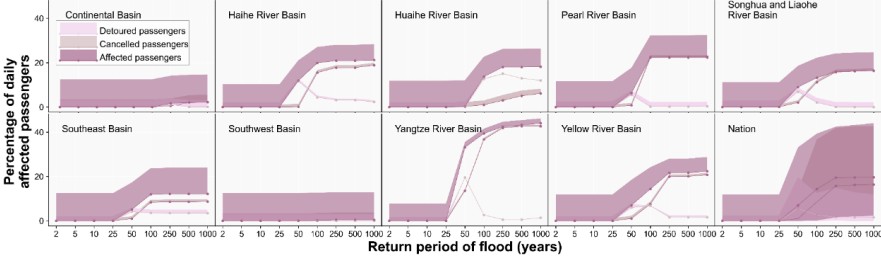

*Fig. A5 system-vulnerability curves of passenger's metrics*
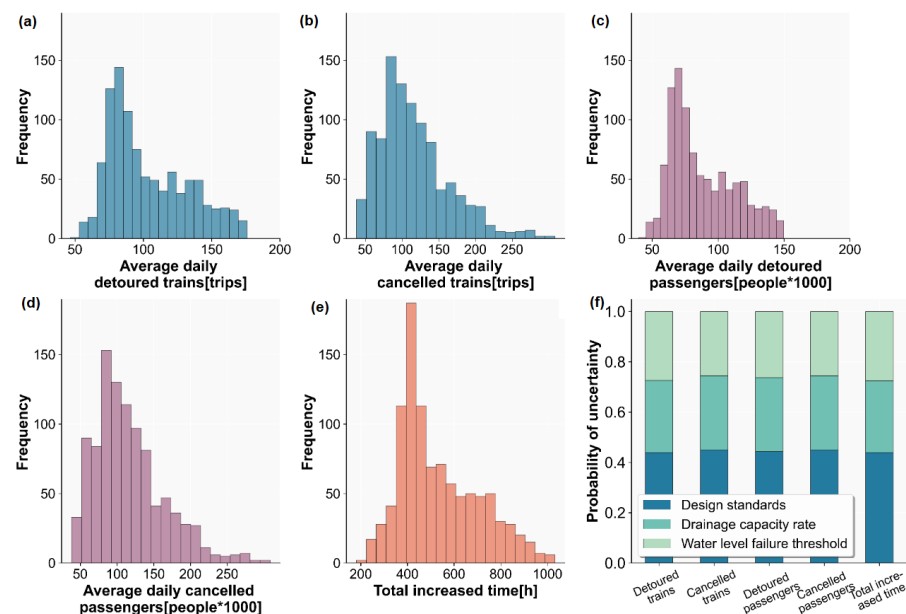

Fig. A6 Results of the uncertainty and sensitivity analyses for the performance metrics.

(a) average daily detoured trains; (b) average daily cancelled trains; (c) average daily

detoured passengers; (d) average daily cancelled passengers; (e) total increased time;

(f) the sensitivity results.




1    Table A1 List of variables

| List of variables | |
|---|---|
| Variable | Description |
| $T$ | Return period of $T$-year |
| $D_T$ | The flood depth with return period of $T$-year |
| $g_{x,y}$ | A grid cell with longitude $x$ and latitude $y$ |
| $D_{T_{x,y}}$ | The flood depth of a flood event of grid cell $g_{x,y}$ with return period of $T$-year |
| $P(D_T)$ | The annual exceedance probability of flood depth $D_T$ |
| $Pr(D_T)$ | A quadratic, continuously differentiable function of $P(D_T)$ |
| $Pr_{x,y}(D_T)$ | A set of continuous inundation depth-exceedance probability functions for $g_{x,y}$ |
| $a, b, c$ | Constant parameters in function $Pr_{x,y}(D_T)$ |
| $B_j$ | River basin $j$ |
| $E_j^i$ | Flood event $i$ in river basin $B_j$ |
| $P_j^i$ | A random number between 0 and 1 for flood event $E_j^i$ in basin $B_j$ |
| $Wd$ | The failure threshold of the railway service after drainage, default value is 0.2 |
| $WL_{x,y}$ | The water level after drainage of grid cell $g_{x,y}$ |
| $Wld_{x,y}$ | The water level of the flood depth under design standard of grid cell $g_{x,y}$ |
| $Dc$ | The drainage capacity rate of Chinese railway system, default value is 0.8 |
| $Z(xy)$ | The failure condition of grid cell $g_{x,y}$ |
| $l_{ij}$ | Rail segment between station $i$ and station $j$ |
| $Fc_{ij}$ | Failure condition of component $l_{ij}$ |
| $FC_{ij}^e$ | The failure condition of railway segment $l_{ij}$ under flood event $e$ |
| $AF_{ij}$ | The annual failure probability of rail segment $l_{ij}$ |
| $E$ | The N-year flood events catalogue |
| $N_S$ | The original number of trains in the system |
| $N_e^s$ | The number of running trains in the system after a flood event |
| $N_e^{tol}$ | The number of daily affected trains under flood event $e$ |
| $N_e^c$ | The number of daily is cancelled trains under flood event $e$ |
| $N_e^d$ | The number of daily detoured trains under flood event $e$ |
| $CA_i$ | The capacity of the $ith$ train |
| $P_e^{tol}$ | The number of affected passengers |
| $P_e^c$ | The number of daily passengers is cancelled passengers under flood event $e$ |
| $P_e^d$ | The number of daily passengers is detoured passengers under flood event $e$ |
| $T_i$ | The original travelling time of the $ith$ train. |
| $T_i^e$ | The running time of the $ith$ train under flood event $e$ |
| $T_e^{tol}$ | The total increased time for detoured trains under flood event $e$ |
| $T_e^{ave}$ | The average increased time under flood event $e$ |
| $AR_s$ | The expected daily flood risk level to the railway system |
| $V_e$ | Performance loss metric, including $N_e^d$, $N_e^c$, $N_e^{tol}$, $P_e^d$, $P_e^c$, $P_e^{tol}$, $T_e^{tol}$, and $T_e^{ave}$ |




1      Table A2 List of all assumptions taken in this study and their range in the sensitivity analysis

| List of all assumptions taken in this study and their range in the sensitivity analysis | | |
|---|---|---|
| Varying parameter | Default values | Range |
| water level failure threshold | 0.2 | [0.1m, 0.5m] |
| drainage capacity rate | 0.8 | [0.7, 0.9] |
| design standards | 100 | [50, 100] |

