# Peer review of "System vulnerability and risk assessment of railway systems to flood events"

_Natural Hazards and Earth System Sciences, 2021_

## Referee Comment (RC2)

[revised manuscript text omitted]

* * *
[1] Geographic Data Sharing Infrastructure, College of Urban and Environmental Science, Peking University (http://geodata.pku.edu.cn).

fit a quadratic spline function to develop an inundation depth-exceedance probability function ($p$) for each grid cell (Marsden, 1974; Vandebogert, 2017; Meshram et al.,

2018). The quadratic spline is a method that uses a piecewise quadratic function to obtain the best-fitting curves. This interpolation method allows us to obtain a smooth continuous curve through the provided flood depths for the different return periods.

The method is applied as follows. For each grid cell, the annual exceedance probability flood depth $D_T$ is calculated by Eq. 1:

$$P(D_T) = \frac{1}{T} \tag{1}$$

where $D_T$ is the magnitude of a flood depth with return period of $T$- year, $P(D_T)$

is the exceedance probability of $D_T$. $D_T$ is between $[D_1, D_{1000}]$, with $D_1 = D_2 \leq$

$D_5 ... \leq D_{1000}$. We assume that $D_1$ is equal to zero (i.e., 1-year event with a flood depth of 0 m) and is the same as that of a 2-year event (the lowest return period in the

GLOFRIS dataset). Let $Pr(D_T)$ denote a quadratic, continuously differentiable function of $P(D_T)$. Then, by definition:

$$Pr(D_T) = aD_T{}^2 + bD_T + c \tag{2}$$

For each  grid cell $g_{x,y}$, we can obtain its piecewise quadratic function by

Eq. 3:

$$Pr_{x,y}(D_T) = \begin{cases} Pr_{x,y}^1(D_T) = a_1 D_T{}^2 + b_1 D_T + c_1 & D_T \epsilon [D_1, D_{1-T}] \\ Pr_{x,y}^2(D_T) = a_2 D_T{}^2 + b_2 D_T + c_2 & D_T \epsilon [D_{1-T}, D_{2-T}] \\ \qquad\qquad \cdots \\ Pr_{x,y}^p(D_T) = a_p D_T{}^2 + b_p D_T + c_p & D_T \epsilon [D_{p-1-T}, D_{p-T}] \end{cases} \tag{3}$$

where $Pr_{x,y}(D_T)$ is a set of continuous inundation depth-exceedance probability functions    consisting    of    $p$    continuous    quadratic    functions.    For

$a(a_1, a_2, \ldots, a_p), b(b_1, b_2, \ldots, b_p), c(c_1, c_2, \ldots, c_p) \epsilon R$, we can calculate these constants by bracketing the critical point of $P(D_T)$ and derivative of the function $Pr_{x,y}(D_T)$; details on the interpolation methods can be found in a previous study by Sun and Yuan (2006). We assume that only one event occurs per year in each basin. Examples of the inundation depth-exceedance probability function of grid cells are shown in Fig. 2a.

**2.1.3 Simulation procedure**

To produce a time-series of flood events based on the created inundation depthexceedance probability functions (Section 2.1.2), we use a Monte Carlo sampling method. The basic idea of the Monte Carlo sampling method is that when the number of simulations is sufficiently large, the frequency of an event approximates the probability of the occurrence of the event (Baker, 2008; Speight et al., 2017). The flood event generation procedure is presented in Fig. 2 and Appendix Fig. A. 1 and can be summarized in two steps. First, we generate a set of independent events at the basin scale. For each event $E_j^i$, and for each basin $B_j$, a random number $P_j^i$ between 0 and is generated from a uniform distribution. The flood depth of the cells in basin  for event $E_j^i$ can be calculated using $P_j^i$ and the inundation depth-exceedance probability function based on the assumption that a flood event in one basin will produce a flood with the same intensity. This concept is presented in Fig. 2a-b. Second, we generate a set of national-scale independent flood events. For a national-scale flood event, basin-specific floods of nine basins can be randomly combined into a nationalscale flood by assuming independence between the flood events among different basins, as presented in Fig. 2c-d.

[Figure]

a) Fit inundation depth-exceedance probability functions
b) Pull the flood depths into basin cells of $B_1$
c) Generate basin-specific flood events

[revised manuscript text omitted]

College of Urban and Environmental Science, P. U. (http://geodata. pku. edu. cn).: Geographic Data Sharing

Infrastructure, n.d.

CRPH: High-speed railway emergency response plan., 2012.

CRPH: Code for design of railway earth structure, TB10001, 2016.

[revised manuscript text omitted]

---

## Author Comment (AC1)

We thank the reviewer for the insightful comments and detailed suggestions on how to improve the manuscript. We found the comments to be very helpful and have incorporated them into the revised manuscript. In the following, the texts with blue font are the reviewer's original comments, the texts with normal font are authors' responses and the texts with italic font are authors' responses in the revised manuscript. Our detailed responses are as follows:

1)   As to Title, "System vulnerability and risk assessment of railway systems to flooding" is failed to reflect the characteristics and innovation of the full text, it is suggested to change "System vulnerability and risk assessment of railway systems to flood events based on national and river basin scale in China".

**Response**: We thank for the reviewer's comments. In the revised manuscript, we have changed the title into "System vulnerability and risk assessment of railway systems to flood events based on national and river basin scale in China"

2)   Part 2, the data used in this paper is not clear, they should be listed in one table or more one.

**Response**: We thank the reviewer for the suggestion. In the revised manuscript, we have added a data table in the appendix document. The added data table is as follows:

*Table A1 List of data*

| Data | Sources |
|---|---|
| *GLOFRIS global fluvial flood hazard* | *Ward et al., 2013; Winsemius et al., 2013 (https://datacatalog.worldbank.org/search/dataset/0038584)* |
| *River basin map* | *http://www.resdc.cn/* |
| *Geographic railway system* | *OpenStreetMap (OSM) (https://www.openstreetmap.org/)* |
| *Train timetable data* | *Chinese Railway Service Website (https://www.12306.cn/index/)* |

3)   In discussion, only text is listed in this part, it is suggested to make a comparative analysis by subject in order to strengthen the practicability and expansibility of the proposed method and framework.

**Response**: We thank the reviewer for the suggestion. In the revised manuscript, we have made a comparative analysis in the discussion part to strengthen the practicability and expansibility of the proposed method and framework. We have added the following description in lines 3-19, page 34 and lines 1-3, page 35:

*In the broader context of risk assessments for transportation systems, the simplified method for generating independent flood events offers a practical method for the large-scale assessment of performance losses and indirect risk. Most existing studies used regional- or national-scale flood footprints to assess flood-induced risk. However, in reality the floods shown in such a flood footprint would not all happen at the same time. For comparison, we calculated the performance loss for the Chinese railway system using national-scale flood footprints (2, 5, 10, 25, 50, 100, 250, 500 and 1000 years in whole China) as shown in Fig. 9 a) and b). Results show that the performance loss for both affected train trips and passengers is almost unaffected for national-scale flood footprints with a return period below 25-years. However, performance loss sharply increases when the flood hazard return period exceeds 50-years. More than 90% of trains and passengers would be affected when the flood hazard return period exceeds 100-years. Compared with the performance loss obtained using the generated independent flood events, the results using the national-scale flood footprints are underestimated for small intensity flood events and overestimated for large intensity flood events. Therefore, when assessing possible cascading effects, the use of independent flood events is necessary (Nones and Pescaroli, 2016).*

[Figure]

*Fig. 9 performance loss for the Chinese railway system using national-scale flood footprints (a) daily affected trains and (b) daily affected passengers*

---

## Author Comment (AC3)

We thank the reviewer for the insightful comments and valuable remarks that helped us to improve the manuscript. The comments are very helpful and we have incorporated them into the revised manuscript. In the following, the texts with blue font are the reviewer's original comments, the texts with normal font are authors' responses and the texts with italic font are authors' responses in the revised manuscript. Our detailed responses are as follows:

1) In 2.1.2, I can't figure out 'why only one event occurs per year in each basin', please describe it more clearly.

**Response**: We apologize for the unclear description and thanks for your suggestions. In this work, we assume that only one event occurs per year in each basin since the intensity of events is equal to or larger than 1-year. In the revised manuscript, we have rewritten Sect 2.1.2 in line 20, page 11 and line 1, page 12.

*In this work, we assume that only one event occurs per year in each basin since we assume the intensity of events is equal to or larger than 1-year.*

2) In 2.3, in a railway system, different railways levels do exist and are designed differently in accordance with the hazard map. Why in this work use the same failure threshold to present it?

**Response**: We thank you for your suggestions. In China, the evidence is that the bridges and embankments for the first-class railway (Backbone railway and quasi-high-speed railway) and second-class railway (Secondary backbone railways and connecting lines), a flood designed protection standard of 100-year return period is used (CRPH, 2016). And in China, since the end of the 21st century, most of the third-class railways (local railway) have been updated to first-class. In addition, the work has analyzed how the risk is influenced by varying the values used for the vulnerability parameters. These results are shown in the sensitivity and uncertainty part of the manuscript in the Sect 3.4 in lines 11-20, page 30, page 31 and lines 1-5, page 32.

3) I think the results in the Appendix need more description.

**Response**: We thank for your suggestions. In the revised manuscript, we have added more description in the article in lines 1-3, page 10 and lines 10-19, page 18 and lines 1-9, page 28 and lines 11-20, page 30 and lines 7-16, page 31.

---

## Author Response (AR1)

Reply on RC1:

We thank the reviewer for the insightful comments and detailed suggestions on how to improve the manuscript. We found the comments to be very helpful and have incorporated them into the revised manuscript. In the following, the texts with blue font are the reviewer's original comments, the texts with normal font are authors' responses and the texts with italic font are authors' responses in the revised manuscript. Our detailed responses are as follows:

1)    As to Title, "System vulnerability and risk assessment of railway systems to flooding" is failed to reflect the characteristics and innovation of the full text, it is suggested to change "System vulnerability and risk assessment of railway systems to flood events based on national and river basin scale in China".

**Response**: We thank for the reviewer's comments. In the revised manuscript, we have changed the title into "System vulnerability and risk assessment of railway systems to flood events based on national and river basin scale in China"

2)    Part 2, the data used in this paper is not clear, they should be listed in one table or more one.

**Response**: We thank the reviewer for the suggestion. In the revised manuscript, we have added a data table in the appendix document. The added data table is as follows:

Table A3 List of data

| Data | Sources |
|---|---|
| Flood hazard data | GLOFRIS global fluvial flood hazard (https://datacatalog.worldbank.org/search/dataset/0038584) |
| River basin map | http://www.resdc.cn/ |
| Geographic railway system | OpenStreetMap (OSM) (https://www.openstreetmap.org/) |
| Train timetable data | Chinese Railway Service Website (https://www.12306.cn/index/) |

3)     In discussion, only text is listed in this part, it is suggested to make a comparative analysis by subject in order to strengthen the practicability and expansibility of the proposed method and framework.

**Response**: We thank the reviewer for the suggestion. In the revised manuscript, we have made a comparative analysis in the discussion part to strengthen the practicability and expansibility of the proposed method and framework. We have added the following description in lines 3-21, page 34 and line 1, page 35:

*In the broader context of risk assessments for the transportation systems, the simplified method for generating independent flood events offers a practical method for large-scale performance losses and indirect risk assessment. Most existing studies used regional- or national-scale flood footprints to assess flood induced risk. However, the presented floods in such a flood footprint may not all happen at the same time. For comparison, we calculated the performance loss for the Chinese railway system using national-scale flood footprints (2, 5, 10, 25, 50, 100, 250, 500 and 1000 years in whole China) as shown in Fig. 9 a) and b). Results show that the performance loss for both affected train trips and passengers is almost unaffected for national-scale flood footprints with a return period below 25-years. While performance loss sharply increases when the flood hazard return period exceeds 50-years. More than 90% trains and passengers would be affected when the flood hazard return period exceeds 100-years,. Compared with the performance loss obtained by generated independent flood events, the results given by national-scale flood footprints are underestimated for small intensity flood events and overestimated for large intensity flood events. Therefore, when assessing possible cascading effects, the use of independent flood events is necessary (Nones and Pescaroli, 2016).*

[Figure]

*Fig. 9 performance loss for the Chinese railway system using national-scale flood footprints (a) dailay affected trains and (b) daily affected passengers*

Reply on RC2:

We thank the reviewer's thorough reading of the manuscript and valuable remarks that helped us to improve the manuscript. The comments are very helpful and we have incorporated them into the revised manuscript. In the following, the texts with blue font are the reviewer's original comments, the texts with normal font are authors' responses and the texts with italic red font are authors' responses in the revised manuscript. Our detailed responses are as follows:

1)   Abstract: the discussion on the results is too large and detailed for an abstract while a brief description on the adopted methodology is totally missing;

**Response**: We thank for reviewer's comments and agree with the reviewer that the abstract need a brief description of methodology and streamline the results. In the revised manuscript, we have rewritten the abstract parts to address the concern of the reviewer. Please refer to lines 12-21, page 1 and lines 1-10, page 2:

*Floods have negative effects on the reliable operation of transportation systems. In China alone, floods cause an average of ~1125 hours of railway service disruptions per year. In this study, we present a simulation framework to analyse the system vulnerability and risk of the railway system to floods. To do so, first, we have developed a novel methodology for generating flood events at both the national and river basin scale. Based on flood hazard maps of different return periods, independent flood events are generated using the Monte Carlo sampling method. Combined with the network theory and spatial analysis method, the resulting event set provides the basis for national- and provincial-level railway risk assessments, focusing in particular on train performance loss. Depending on this framework applied for the Chinese railway system, the results show that due to spatial variations in the railway topology and traffic flows, the system vulnerability of the Chinese railway system to floods in different basins is highly heterogeneous. Flood events in the Yangtze River Basin show the largest impact on the national railway system, with approximately 40% of the national daily trains being affected by a 100-year flood event in that basin. At the national level, the average number of daily affected trains and passengers for the national system are approximately 200 trips and 165,000 people (2.7% and 2.8% of the total daily numbers of trips and passengers), respectively. The event-based approach presented in this study shows how we can*

*identify critical hotspots within a complex network, taking the first steps in developing climate-resilient infrastructure.*

2) Introduction: the introduction should provide also some further details on both the adopted methodology and metrics. An anticipation of the analyses that will be carried out is essential to encourage potential readers to go through the paper. Novelty of the proposed approach should be better stressed.

**Response**: We thank for the reviewer's suggestions. In the revised manuscript, we have added more details on both adopted methodology and metrics in the introduction section in lines 14-21, page 5 and lines 1-8, page 6; and the novelty of the proposed approach have been stressed in lines.

*This study aims to develop a framework to quantify the system vulnerability and risk in transportation systems in terms of operational performance loss under large-scale flood hazards. System vulnerability in this study is represented as the system performance loss with different flood intensities. When assessing possible cascading effects, the use of independent flood events is necessary (Nones and Pescaroli, 2016), as the presented floods in regional-or national-scale flood footprints, which show the flood depth for a given return period in that area, may not all happen at the same time. To overcome the shortcomings in existing studies, we develop simplified practicable and novel method for generating a set of independent flood events at the national and river basin scale. The independent floods are generated using a curve fitting method and Monte Carlo sampling method based on global flood hazard model maps and river basins. By coupling simulated flood events with the railway network using the spatial analysis method, we identify the railway failure hotspots caused by floods. At the same time, potential performance loss from trains' and passenger's perspective is assessed using network theory. We illustrate our methodology by applying it to the Chinese railway system.*

3) In Data and Method section and sect 2.1.1, the global flood hazard model should be better described (also providing some examples in the SM). All the adopted metrics should be defined much more carefully, with a more precise and effective use of terms. For instance, only the trains where passenger travel can be cancelled or detoured, while passengers cannot be cancelled or detoured; so the metrics named 'passenger cancelled induce' or 'passenger detoured' in my opinion should be renamed (and better defined at their first appearance in the text).

**Response**: We thank for the reviewer's suggestions. In the revised manuscript, we have added more details on the global flood hazard model and providing the flood maps on 50 and 500-year events in supplement materials in lines 10-20, page 9 and lines 1-4, page 10. Adopted metrics have been redefined and please refer to our response to Question 4.

*Our flood hazard data are extracted from the GLOFRIS global fluvial flood hazard maps of Winsemius et al. (2013), which are developed using the GLOFRIS modelling cascade provided in Ward et al. (2013) and Winsemius et al. (2013). The GLOFRIS modelling cascade first simulates daily discharge using the PCRaster GlobalWater Balance (PCR-GLOBWB) global hydrological model [Beek and Bierkens, 2008; Beek et al., 2011]. Based on daily discharge, daily flood volumes are simulated using the PCR-GLOBWB extension for dynamic routing, DynRout (PCR-GLOBWB-DynRout) [Ward et al., 2013; Winsemius et al., 2013]. In the next step, flood volumes, for different return periods: 2, 5, 10, 25, 50, 100, 250, 500 and 1000 years, are obtained using the annual year time series for maximum flood volumes by fitting a Gumbel distribution. These flood volumes are then converted into inundation maps (30-arcsecond, ca. 1-km) using the inundation downscaling model of GLOFRIS [Winsemius et al., 2013]. In the appendix materials, we provide flood maps of 50 and 500-year events. Maps show that the inundation depth highly varies in China. Railway lines in eastern coastal China and South China are faced with the most severe floods.*

[Figure]

Fig. A1 (a) the 50-year flood, (b) the 500- year flood

4) The description of the fitting procedure (Sect.2.1.2) must be improved. Figure 2a is rather unclear to me, and the caption does not help the readers. Moreover, its size is too small and the inset legend cannot be read (similar problems are present also in Figures 5, 6 and 7). I would suggest to

place the four graph in figure 2 in a 2 x 2 grid, enlarging each graph. Caption must be more clear for figure 2a and more concise for figs 2b, 2c and 2d.

**Response**: We apologize for the unclear description and thanks for your suggestions. In the revised manuscript, we have rewritten Sect 2.1.2 in lines 10-20, page 10, page 11, and lines 1-4, page 12. At the same time, Figure 2a as well as Figures 5, 6, 7 have been improved as followed.

**2.1.2 Fitting procedure**

*For each grid cell, the GLOFRIS maps estimate the flood depth for the nine aforementioned return periods (2, 5, 10, 25, 50, 100, 250, 500 and 1000 years). To estimate the flood depth for any return period, we fit a quadratic spline function to develop an inundation depth-exceedance probability function (P) for each return period interval for each grid cell (Marsden, 1974; Vandebogert, 2017; Meshram et al., 2018). The quadratic spline is a method that uses a piecewise quadratic function to obtain the best-fitting curves. This interpolation method allows us to obtain a smooth continuous curve through the provided flood depths for the different return periods.*

*The method is applied as follows and examples of the inundation depth-exceedance probability function of grid cells are shown in Fig. 2a:*

*For each grid cell $g_{x,y}$, the annual exceedance probability flood depth $D_T$ is calculated by Eq. 1:*

$$P(D_T) = \frac{1}{T} \tag{1}$$

*where $D_T$ is the magnitude of a flood depth with a return period of T- year, $P(D_T)$ is the exceedance probability of $D_T$.*

*Let $Pr(D_T)$ denotes a quadratic, continuously differentiable function of $P(D_T)$. Then, by definition:*

$$Pr(D_T) = aD_T{}^2 + bD_T + c \tag{2}$$

*For each return period interval of grid cell $g_{x,y}$, we can obtain its piecewise quadratic function by Eq. 3:*

$$Pr_{x,y}(D_T) = \begin{cases} Pr^1_{x,y}(D_T) = a_1D_T{}^2 + b_1D_T + c_1 & D_T \epsilon [D_2, D_5] \\ Pr^2_{x,y}(D_T) = a_2D_T{}^2 + b_2D_T + c_2 & D_T \epsilon [D_5, D_{10}] \\ \dots \\ Pr^8_{x,y}(D_T) = a_8D_T{}^2 + b_8D_T + c_8 & D_T \epsilon [D_{500}, D_{1000}] \end{cases} \tag{3}$$

*where $Pr_{x,y}(D_T)$ is a set of continuous inundation depth-exceedance probability functions*

*consisting of 8 continuous quadratic functions for $g_{x,y}$ and shows in Fig. 2a with curves. For*

$a(a_1, a_2, \ldots, a_8), b(b_1, b_2, \ldots, b_8), c(c_1, c_2, \ldots, c_8) \epsilon R$ , *we can calculate these constants by*

*bracketing the critical point of $\mathrm{P}(D_T)$ and derivative of the function $Pr_{x,y}(D_T)$; details on the*

*interpolation methods can be found in a previous study by Sun and Yuan (2006). In this work, we*

*assume that only one event occurs per year in each basin since we assume the intensity of events is*

*equal to or larger than 1-year. When the return period is lower than 2, the flood depth is set to zero*

*which is the same as that of a 2-year event.*

[Figure]

**(a) Fit inundation depth-exceedance probability functions**

**(b) Generate random number $P_i$ for basin $i$ to get independent flood event for each basin and combine them to obtain a national-scale event**

**(c) Repeat process (b) and generate 10000 national-scale flood events**

***Fig. 2 An example of generating national-scale flood events. In (b), p1, p2 , p3 , and p4 are***

***the random number between 0 and 1 generated for basin $B_1, B_2, B_3$ and $B_4$, which are used to***

***generate basin-scale events based on the functions in (a). The layers of basin-scale floods in (b)***

***are combined into a national-scale flood event. The layers in (c) is the 10000 national-scale***

***events using the process in (b).***

**2.1.3 Simulation procedure**

*To produce a time-series of flood events based on the created inundation depth-exceedance*

*probability functions (Section 2.1.2), we use a Monte Carlo sampling method. The basic idea of the*

*Monte Carlo sampling method is that when the number of simulations is sufficiently large, the*

*frequency of an event approximates the probability of the occurrence of the event (Baker, 2008;*

*Speight et al., 2017). The flood event generation procedure is presented in Fig. 2 and Appendix Fig.*

*A1 and can be summarized in two steps. First, we generate independent events at each basin and*

*combined them into a national event. For an event $E_j^i$, and for each basin $B_j$, a random number*

*$P_j^i$ between 0 and 1 is generated from a uniform distribution. The flood depth of the cells in basin*

*$B_j$ for event $E_j^i$ can be calculated using $P_j^i$ and the inundation depth-exceedance probability*

*function based on the assumption that a flood event in one basin will produce a flood with the same intensity. For a national-scale flood event, basin-specific floods of nine basins can be randomly combined into a national-scale flood by assuming independence between the flood events among different basins, this concept is presented in Fig. 2b. Second, we repeat this process for 10000 times to generate a set of national-scale independent flood events as presented in Fig. 2c.*

[Figure]

**Fig. 5 Exceedance probability-performance loss curves**

[Figure]

**Fig. 6 Performance loss of the railway system per province.**

[Figure]

**Fig. 7 System vulnerability curves induced by river floods from the national flood event set**

5) 2.4 could be renamed "performance loss metrics" and restructured with a separate subsection for each metric. Subsection 2.4.2 could become sect. 2.5. All the assumptions made for the metrics definition must be better clarified.

**Response**: We thank for the reviewer's suggestions. In the revised manuscript, we renamed

"performance loss metrics" and restructured them in a separate subsection. For the assumptions of the metrics clearer, we added some descriptions in lines 2-4, page 18:

*2.4 Performance loss metrics*

  *2.4.1 Daily affected trains and passengers*

  *2.4.2 Daily detoured trains and passengers influenced by detoured train*

  *2.4.3 Total increased time for the detoured trains*

  *2.4.4 Average increased time for the detoured trains*

  *2.4.5 Daily cancelled trains and passengers influenced by cancelled train*

*2.5 Calculating system vulnerability and risk*

*2.6 2.6 Uncertainty and sensitivity analysis*

  *We assume that the average number of passengers is 80% of the train's capacity (Wei et al., 2017; Rezvani et al., 2015). As such, the number of affected passengers $P_e^{tol}$ can be defined by Eq. 9:*

6) Results section presents a quite good description of the results while comments on the potential implications of the various results are almost totally missing or present only in the discussion section; this aspect could be improved. The discussion on the results of the sensitivity and uncertainty analysis in Setc.3.4 should be considerably improved; for instance, pie charts in Fig.8 should be explained and commented.

**Response**: We thank for the reviewer's suggestions. In the revised manuscript, we add the comments on the potential implications of the various results in results parts in lines 14-17, page 22, lines 1-4, page 27, and lines 7-9, page 30 and lines 6-15, page 31. Meantime, the results of the sensitivity and uncertainty also improved.

**3.1 Failure hotspots of railway segments**

[revised manuscript text omitted]

*uncertainty and improve the performance loss estimates.*

7) Some practical examples of the utility of the proposed approach should be reported in the conclusion to highlight the importance of the work.

**Response**: We thank for the reviewer's suggestions. In the revised manuscript, we have added some practical examples of the utility of the proposed approach in the conclusion part page 20 and lines 2-17, page 37.

*The developed system vulnerability curves and flood risk maps can provide the information for the decisions on safety and effectiveness of operation and maintenance. Various performance metrics can be considered by management departments based on their particular problems. Using our current approach, the performance loss can be used as the start of the indirect risk assessment from the travel journey perspective. By combining the ticket prices and the operating cost per kilometre, the economic loss for the railway company can be calculated based on the affected trains and associated passengers (Lamb et al., 2019). As a key mode of transport for interregional trade, the failure of railway systems can produce large shocks for industries dependent on the supply that may come from flooded businesses. The risk values per province (such as expected daily cancelled trains and passengers) can be used as indicators to link with business disruptions. Future work can try to assess the interregional trade based on the Input and Output table and regional railway transportation performance decreased in this work. The assessment of shocks and indirect economic losses induced by railway system failures is essential for policymakers to design railway infrastructures and to measure indirect economic losses.*

8) Please double-check your References. I have found out some inconsistencies. For example, in your manuscript you refer to Liu 2018a and 2018b, but in the References I have found Liu 2009, Liu 2018 and Lyu 2018.

**Response**: We thank you for the reviewer's suggestions. We have checked the references and revised them in the revised manuscript.

Reply on CC1:

We thank the reviewer for the insightful comments and valuable remarks that helped us to improve the manuscript. The comments are very helpful and we have incorporated them into the revised manuscript. In the following, the texts with blue font are the reviewer's original comments, the texts with normal font are authors' responses and the texts with italic font are authors' responses in the revised manuscript. Our detailed responses are as follows:

1)  In 2.1.2, I can't figure out 'why only one event occurs per year in each basin', please describe it more clearly.

**Response**: We apologize for the unclear description and thanks for your suggestions. In this work, we assume that only one event occurs per year in each basin since the intensity of events is equal to or larger than 1-year. In the revised manuscript, we have rewritten Sect 2.1.2 in lines, page 20 and lines 1-4, page 12.

*In this work, we assume that only one event occurs per year in each basin since we assume the intensity of events is equal to or larger than 1-year.*

2)  In 2.3, in a railway system, different railways levels do exist and are designed differently in accordance with the hazard map. Why in this work use the same failure threshold to present it?

**Response**: We thank for your suggestions.   In China, the evidence is that for the first-class railway, flood protection is a 100-year flood standard referred from (CRPH, 2016). And in China, after the 21st century, most of the second-class and third-class were updated to first-class. In addition, the work has analyzed the risk influence vary the vulnerability parameters in the sensitivity and uncertainty part. I think it is suitable for the assumption.

3)  I think the results in the Appendix need more description.

**Response**: We thank for your suggestions. In the revised manuscript, we have added more description in the article in lines 10-19, page 18 and lines 1-9, page 28 and in lines 11-20, page 30 and lines 6-15, page 31.

---

## Author Response (AR2)

Reply on RC1:

We thank the reviewer for the insightful comments and detailed suggestions on how to improve the manuscript. We found the comments to be very helpful and have incorporated them into the revised manuscript. In the following, the texts with blue font are the reviewer's original comments, the texts with normal font are authors' responses and the texts with italic font are authors' responses in the revised manuscript. Our detailed responses are as follows:

1) The structure and table of contents of the article need to be adjusted to make the overall presentation of the article clearer. For example, the first part is the introduction, and the second part is the data and methods. It is suggested to list 2.1 data sources, 2.2 methods and processing procedures. The third part is the result analysis. the fourth part is the discussion and the fifth part is the conclusion.

Response: We thank for the reviewer's comments. In the revised manuscript, we have adjusted the structure and table of the contents of the articles as the reviewer suggested. The first part is the introduction. The second is the data and method, 2.1 data sources, 2.2 methods and processing procedures. The third part is the result. The fourth part is the discussion, and the fifth part is the conclusion. Please refer to the structures and the content in the revised manuscript.

2) The conclusion of the article needs to be sorted out again to make it clearer. It is suggested to list 1,2,3 and so on.

**Response**: We thank the reviewer for the suggestion. In the revised manuscript, we have used the 1,2,3 to resort the conclusion. Please refer to lines 4-16, page 36, page 37 and lines 1-5, page 38:

*5    Conclusion*

*The increased frequency of extreme flood events, coupled with interregional trade growth, requires national- and global-scale transportation networks to be more resilient to cope with disruptive events. Evaluation of system-level vulnerability and identification of risk hotspots is a first step to enhance the robustness of the transport system. This study presents a framework for performing system-level vulnerability and risk assessments of a railway system under flooding. The developed framework couples simulated flood events with state-of-the-art network analysis to measure system disruptions caused by floods to identify risk hotspots. The system vulnerability and*

*risk induced by the flooding are quantified in terms of the performance loss of the Chinese railway system. Results show that failure hotspots, system vulnerability and the risk of the Chinese railway system under floods are highly heterogeneous. The main conclusions are as follows:*

1. *High failure hotspots are mainly distributed in South China, i.e. Yangtze River, Pearl River and Southeast Basins. In addition, floods in the basins in central and eastern China have the highest impacts on the Chinese railway system. Floods in the Yangtze River Basin have the largest impact on the daily cancelled trains. At the same time, floods in the Huaihe and Haihe River Basins cause the largest number of detoured trains as well as associated increased time for the Chinese railway system compared with other basins.*

2. *At a national level, the average percentage of daily affected trains and passengers for the national system is approximately 2.7%. The mean average increased time for detoured trains reaches approximately 5 hours. At the provincial level, the provinces in Central China have the highest risks, estimated to be 4.5% relative to the number of the province's daily trains and more than 3.5% relative to the number of the province's daily passengers. The high risk in terms of the total increased time is mostly distributed in East China, whereas the highest average increased time is distributed in western provinces, such as Xinjiang and Tibet Provinces.*

3. *Using our current approach, the performance loss can be used as the start of the indirect risk assessment from the travel journey perspective. By combining the ticket prices and the operating cost per kilometre, the economic loss for the railway company can be calculated based on the affected trains and associated passengers (Lamb et al., 2019). As a key mode of transport for interregional trade, the failure of railway systems can produce large shocks for industries that depend on the supply that may come from flooded businesses. The risk values per province (such as expected daily cancelled trains) can be used as indicators to link with business disruptions. Future work can try to assess the shocks and indirect economic losses based on the Input and Output table and regional railway transportation performance decreased in our work.*

Reply on RC2:

We apologize for neglecting the comments that the reviewer made on the annotated paper. In the second round revision, these comments are carefully considered and incorporated in the revised paper. In addition, we thank the reviewer's thorough reading of the manuscript and valuable remarks that helped us to improve the manuscript. In the following, the texts with blue font are the reviewer's original comments, the texts with normal font are authors' responses, and the texts with italic red font are authors' responses in the revised manuscript. Our detailed responses are as follows:

1) Try to report only the main results in percentage terms (I would suggest to avoid to discuss results in absolute values in the abstract section)

**Response**: We thank the reviewer for the suggestion. In the revised manuscript, we have described the main results in percentage terms in the abstract section. We have added the following description in lines 5-7, page 1:

*At the national level, the average percentage of daily affected trains and passengers for the national system is approximately 2.7% of the total daily numbers of trips and passengers.*

2) try to add some keywords not contained into the title

**Response**: We thank the reviewer for the suggestion. We have added " Monte Carlo method " and " network analysis" as the keywords. Please refer to lines 10-11, page 1:

*KEYWORDS: railway system; flood; risk assessment; system vulnerability; Monte Carlo method; network analysis*

3) please, explain what you exactly mean by "direct" economic loss

**Response**: "Direct economic loss" refers to "the costs for repairing the damaged railway infrastructure". We have added this description in lines 6-7, page 3:

*In 2016, the direct economic loss (i.e., the costs for repairing the damaged railway infrastructure) of the Chinese railway system caused by floods was approximately 80 million USD (Editorial Board of China Railway Yearbook, 2001-2017).*

4) Informal. I would suggest something like: as a result, consequently, therefore, etc.

**Response**: We thank the reviewer for the suggestion. We have changed the "such as" word into "as a result", "consequently", "therefore", etc throughout the manuscript to make the sentence more formal.

5) *seasonal variations and feedback dynamics problem*

➢ *I can image that you have assumed a timetable constant over time, without considering seasonal variations in the frequency of trips, new routes, etc? Is this correct? How realistic is this assumption and how potential seasonal variations can affect the results?*

➢ *also in this case, you didn't account for any potential seasonal variations? Could this affect the results? Of course your goal is to analyse the average number of affected trains and passengers over the year, but it could be usefull to analyse the typical period of occurrence of the main floods with respect to the seasonal variability of the train trips and number of passengers. Moreover, often the cancellation of one or more trains could imply an increment in the number of passengers in other trains that offer an alternative way to the cancelled trip. Did you account for any possible feedback dynamics on the number of passengers in the case of train cancellation?*

➢ *Probably the analysis of the results considering the seasonality in flood events occurrence in relation to potential seasonal variations in the timetable (number of trains) and passengers deserves to be explored in future and mentioned here.*

**Response**: We thank the reviewer for the comments and agree that the performance loss is affected by the seasonal variations of the trip timetable and feedback dynamics problem. Due to lacking timetable and passengers' capacity day by day, it is difficult to consider seasonal variations of the trip timetable and feedback dynamics problem. We have added the following description in the discussion part in the revised manuscript to address the problem. Please refer to lines 13-19, page 34:

*Due to lacking timetable and passengers' capacity day by day, we have assumed a timetable constant over time, without considering potential seasonal variations as well any possible feedback dynamics on the number of passengers in the case of train cancellation. Since our goal is to analyse the average number of affected trains and passengers over the year, the assumption is reasonable. In future work, it is worth investigating the typical period of occurrence of the main floods concerning*

*the seasonal variability of the train trips and the number of passengers.*

6) This figure has been recalled in the manuscript also to show the main basins in which has been divided the China (page 9 - line 11). Please add this information also in the caption.

**Response**: We thank the reviewer for the suggestion. We have added main basins information in the caption: The spatial distribution of the railway network, average daily numbers of trains and the main river basin in China.

7) The authors are right but I think that they should provide some more details on what they mean by intensity. In some cases, the hazard is defined only on the base of the water depth, while in some other cases it can take into account also the combination of depth and velocity.

**Response**: We thank the reviewer for the suggestion. In this work, the hazard is defined on the base of the water depth (m). In the revised manuscript, we have added the following description in lines 16-17, page 9:

*In this work, the hazard intensity is represented by the water depth (m).*

8) Here you should clearly define the aspects related to the design standard under consideration (i.e. water level for cluverts, bridges, embankments, etc,) and the type of threshold selected (i.e. water level) with unit (m).

**Response**: We thank the reviewer for the suggestion. We have addressed this in Section 2.2.3. Please refer to lines 4-7, page 18:

*The flood design standard of the culverts, bridges and embankments of the Chinese national railway system is designed for 100-year water depth, according to the standard for flood control (CRPH, 2016).*

9) Much of the information in Section "2.1 National-scale flood event generation" are already reported in Section "2. Data and method". It is for this reason that I would suggest authors to remove this section and move (and integrate) information to section 2.

**Response**: We thank the reviewer for the suggestion. In the revised manuscript, we have removed most information of this part and moved (and integrated) information to section 2. Please refer to

lines 6-21, page 10 and lines 6-11, page 11:

*Figure 2 presents an overview of the framework used in this study. First, we generate a national- and river basin-scale flood event set. To do this, we use flood hazard maps for different return periods at the national scale, taken from a global flood hazard model (see Section 2.1.1). We then divide these into flood hazard maps for the major river basins and use a curve-fitting method to estimate the flood depth for any return period for any cell. We then apply a Monte Carlo sampling method (Metropolis 1987) to generate the flood events per river basin and aggregate these events to the national scale. Second, we define the railway system as a network using network theory (Newman, 2010). Third, we intersect the flood events with the railway network to identify the disrupted segments in the railway system based on a pre-defined failure threshold. In the last part of our analysis, we assess the system vulnerability and risk in terms of several performance loss metrics, including the daily total number and the total percentage of trains affected (i.e. cancelled or detoured) and involved passengers as well as the total increased time and the average increased time for the detoured trains. We also analyse the parameters sensitivity in the failure threshold and the related risk uncertainty.*

**2.2.1 National-scale flood event generation**

*To ensure the estimation is as accurate as possible for an event-based flood risk assessment based on the Monte Carlo sampling, a large number of independent flood events are required (Speight et al., 2017; Wu, 2019; Zhu et al., 2020). In the following subsections, we will describe the procedures to generate flood events, including input flood hazard maps, the function fitting procedure, and the Monte Carlo analysis in more detail.*

10) Please, explain this sentence. Is this related to the adopted Monte Carlo sampling method or a more general statement? In this last case, why you should need of a large number of independent flood events to make an accurate flood risk assessment? Maybe, in order to characterise the return time of the flood events, or what else?

**Response**: The basic idea of the Monte Carlo sampling method is that when the number of simulations is sufficiently large, the frequency of an event approximates the probability of the occurrence of the event. Therefore, a large number of independent flood events are required to ensure the estimation is as accurate as possible for an event-based flood risk assessment based on

the Monte Carlo sampling. To make it clearer, In the revised manuscript, we have added the following description in lines 6-8, page 11:

*To ensure the estimation is as accurate as possible for an event-based flood risk assessment based on the Monte Carlo sampling, a large number of independent flood events are required (Speight et al., 2017; Wu, 2019; Zhu et al., 2020)*

11) I guess that x, y are respectively the horizontal and vertical coordinate of any cell center. Is that correct? Please specify.

**Response**: It is correct, x, y are respectively the horizontal and vertical coordinate of any cell center. In the revised manuscript, we have added the following description in lines 15-17, page 12:

*For each grid cell $g_{x,y}$ ($x$, $y$ are respectively the horizontal and vertical coordinate of grid cell centre), the annual exceedance probability flood depth $D_T$ is calculated by Eq. 1:*

12) This is rather unclear to me. Why you need to define a T=1 year event, why you are associating a null corresponding depth (0 m) to a 1-year event? Why is it the same as that of 2-year event? Why did you decide to consider also 1-year return period? I add that talking about T=1 year event does not make much sense to me. It means P = 1 (see eq. 1) and (I think) that's why they impose the 1-year event equal to 0.

**Response**: (1) For the question:" Why you need to define a T=1 year event? Why did you decide to consider also a 1-year return period? I add that talking about T=1 year event does not make much sense to me. It means P = 1 (see eq. 1) and (I think) that's why they impose the 1-year event equal to 0." It is considering that the inundation depth-exceedance probability is from 0 to 1, when T=2 year event, the P=1/2; and when T=5, the P=1/5, when we impose the 1-year event, the P=1, which can make the inundation depth-exceedance probability is from 0 to 1. (2) For the question: "why you are associating a null corresponding depth (0 m) to a 1-year event? Why is it the same as that of 2-year event? why they impose the 1-year event equal to 0." It is because the depth of the 2-year event in GLOFRIS global fluvial flood hazard maps is equal to 0 m. So we assume that $D_1$ is equal to zero (i.e., 1-year event with a flood depth of 0 m). In the revised manuscript, we have added the following description into footnote in page 13:

*It is considering that the inundation depth-exceedance probability is from 0 to 1, when T=2-year*

*event, the P=1/2; and when T=5-year event, the P=1/5, when we impose the 1-year event, the P=1, which can make the inundation depth-exceedance probability is from 0 to 1.*

*As the depth of the 2-year event in GLOFRIS global fluvial flood hazard maps is equal to 0 m. We assume $D_1$ is also equal to zero (i.e., 1-year event with a flood depth of 0 m).*

13) Please specify what i and j represent and their range.

**Response**: We thank the reviewer for the suggestion. In the revised manuscript, we have added the following description in lines 11-13, page 14:

*"$i$" is the sequence number of simulated flood event; "$j$" is the sequence number of basin number, which belongs to (1,9)*

14) I guess this is referred to the return period. Please rewrite.

**Response**: We thank the reviewer for the suggestion. In the revised manuscript, we have rewritten 100 year into 100-year event, etc, throughout the manuscript.

15) here, you should use 'Netot' instead of 'Nec+Ned'

**Response**: In the revised manuscript, we have changed $N_e^c + N_e^d$ into $N_e^{tol}$.

$$P_e^{tol} = \sum_i^{N_e^{tol}} CA_i * 0.8$$

16) what does the drainage capacity rate take into account? Why you have assumed the value 0.8? This didn't come from sensitivity analysis, right?

**Response**: The value and the concept of the drainage capacity rate are referred to Espinet et al., (2018), which is defined as the drainage capacity of embankment, bridge and culvert. In this work, the value is 0.7 for bridges and culverts in Mozambique. Considering China is more developed than Mozambique, we assume the infrastructure in China has a higher drainage capacity. In addition, we do also think that the parameter will lead to a large uncertainty to the performance loss. Therefore, we perform a sensitivity and uncertainty analysis in Section 3.4. In the revised manuscript, we have added the following description into footnote in page 18:

*The value and the concept of the drainage capacity rate are referred to Espinet et al., (2018), which*

*is defined as the drainage capacity of embankment, bridge and culvert. In this work, the value is 0.7 for bridges and culverts in Mozambique. Considering China is more developed than Mozambique, we assume the infrastructure in China has a higher drainage capacity and a value of 0.8 is assigned.*

17) What values have you considered for CA. What is the source? Probably this information should stay in section 2.2.

**Response**: We thank the reviewer for the suggestion. In the revised manuscript, we have added the CA information in Section 2.1.3. Please refer to lines 4-12, page 8:

*2.1.3 Chinese Railway data*

*The geographic information, time table data, as well passenger's capacity data of Chinese railway are collected. The geographic information of railway system from OpenStreetMap (OSM), which provide the spatial distribution of the Chinese railway system (Fig. 1). The timetable data, which includes the daily number of trains and associated routes from the Railway Service Website, while the passenger's capacity data is obtained from https://www.china-emu.cn/ for China High-Speed Train (G Train, D Train, and C Train) and https://zh.wikipedia.org/wiki for others (Z Train, T Train, K Train, etc.).*

18) Is this 'increased time' limit supported by any directive or something else?

**Response**: In China, the operation cycle of each train is once a day. Therefore, if the increased time for re-routing one trip is greater than 24 hours, it will stop and the new trip will be started the next day.

19) I would suggest to use the same background map as in the Figure 3. This would make possible to make a clearer distinction in terms of basins as well. Also, I suggest to specify that the bar named National is referred to the entire domain.

**Response**: We thank the reviewer for the suggestion. In the revised manuscript, we have changed the background map for Fig. 4 and changed the National to the entire domain. Please refer to line 14, page 23:

[Figure]

20) This is not true for the Soutwest and Continenatal basins, where the percentage are pretty much constant. Does the author investigate this aspect?

**Response**: The low and constant impacts of daily affected trains observed in the southwest and continental basins are due to a lower railway line density and daily train flows. At the same time, the lower annual failure probability of the rail segments in these areas also leads to a lower probability of failed railway segments per flood event and results in lower impact. In the manuscript, we have added explanations in lines 9-21, page 29 and line 1, page 30:

*For most basins, between the 25-year and 100-year flood events, the percentage of daily affected trains and daily cancelled trains relative to the total number of daily trains per flood event increases. The rule is not suitable for the Soutwest and Continenatal basins, where the percentage are pretty much constant and low. It is due to a lower railway line density and train trips in these two basins. A low impact is expected even all railway lines are disrupted. While the percentage of daily detoured trains relative to total daily trains and the total and average increased time, increases between the 25-year and 50-year flood events, and sharply decreases between the 50-year and 100-year events, especially for the Yangtze River, Yellow River and Pear River Basin floods. This is because most of the north-south rail lines in China, such as the Beijing-Guangzhou and Beijing-Jiulong lines, cross these basins. Most trains that are detoured under a 50-year event cannot be detoured under a 100-year event, as most of the north-south rail lines suffer failures at this hazard intensity.*

21) Could you report this map in the SM?

**Response**: We thank the reviewer for the suggestion. In the revised manuscript, we have added this

map in the SM. Please refer to lines 1-3, page 50

[Figure]

*Fig. A8 Susceptibility map of the national railway network subjected to flood (source: Liu et al.,*

*2018b)*